# Influence of the chirality of carbon nanodots on their interaction with proteins and cells

Huijie Yan[1,8], Michele Cacioppo [1,2,8], Saad Megahed [1,3], Francesca Arcudi [2], Luka Đorđević [2], Dingcheng Zhu[1,4], Florian Schulz[1], Maurizio Prato [2,5,6✉], Wolfgang J. Parak [1,5✉] & Neus Feliu[1,7✉]

Carbon nanodots with opposite chirality possess the same major physicochemical properties such as optical features, hydrodynamic diameter, and colloidal stability. Here, a detailed analysis about the comparison of the concentration of both carbon nanodots is carried out, putting a threshold to when differences in biological behavior may be related to chirality and may exclude effects based merely on differences in exposure concentrations due to uncertainties in concentration determination. The present study approaches this comparative analysis evaluating two basic biological phenomena, the protein adsorption and cell internalization. We find how a meticulous concentration error estimation enables the evaluation of the differences in biological effects related to chirality.

[1] Fachbereich Physik, Center for Hybrid Nanostructures (CHyN), Universitat Hamburg, 22607 Hamburg, Germany. [2] Department of Chemical and Pharmaceutical Sciences, INSTM UdR Trieste, University of Trieste, Via Licio Giorgieri 1, 34127 Trieste, Italy. [3] Physics Department, Faculty of Science, Al-Azhar University, Cairo, Egypt. [4] College of Material, Chemistry and Chemical Engineering, Hangzhou Normal University, Hangzhou, PR China. [5] Center for Cooperative Research in Biomaterials (CIC biomaGUNE), Basque Research and Technology Alliance (BRTA), Paseo de Miramon 182, 20014 Donostia San Sebastian, Spain. [6] Basque Foundation for Science, Ikerbasque, 48013 Bilbao, Spain. [7] Fraunhofer Center for Applied Nanotechnology (CAN), 20146 Hamburg, Germany. [8] These authors contributed equally: Huijie Yan, Michele Cacioppo. ✉email: mprato@cicbiomagune.es; wolfgang.parak@uni-hamburg.de; neus.feliu@physnet.uni-hamburg.de

While chemically similar, molecular isomers with different chirality can have significant different biological impact, such as pharmaceutical effect or cellular toxicity[1–7]. This has been well investigated on the level of small organic chiral isomers, which are naturally existing or can be synthesized. In the last decades, the concept of chirality in biological interactions has gained ever more interest in the raising world of nanomaterials, such as nanoparticles (NPs) or assemblies of NPs[8–15]. Apart from some atomically defined metal clusters, NPs in general do neither possess a defined molecular formula or structural formula. As thus NPs of one type are not identical but will have a (narrow) size-, shape-, and charge-distribution, the question is if chirality on the size level of whole NPs plays a role with the NPs interaction with biology, such as protein adsorption and cellular uptake. In fact, reports exist in which adsorption of proteins, the so-called protein corona[16] was found to depend on the chirality of the NPs[17–23]. Previous studies on carbon nanodots (CNDs) have focused on studying the effect of opposite chiral carbon dots in their biocompatibility and toxicity to liver HepG2 cells[9], stereoselective interaction with the prion protein[24], tuning enzymatic activity[25–27], interaction with the Golgi apparatus[28], studying their effect on plant growth[29], as well as establishing methods for detecting the interaction between achiral carbon dots and proteins[30].

For a quantitative analysis, there are several complications. As the NPs of one type will not be identical, but there will be a distribution of their properties, a potential effect of NPs of the same type but with different chirality might be at a lower level than the inhomogeneity in effect due to the distribution of NP properties. Also, differences in chirality may involve additional distributions in the NPs properties, such as size, optical properties, colloidal stability, etc., as two types of NPs with different chirality will originate from two distinct batches of synthesis. The paramount requirement for analyzing the effect of chirality on the interaction of NPs with biology thus will be NPs with narrow distributions of their properties. In addition, in order to directly compare the biological impact of NPs with different chirality a metric needs to be defined on how properties of different NPs can be compared at the same concentration. Given the fact that surface coatings modify the molecular weight of NPs[31], it is not the same metric to measure at the same mass concentration or to measure at the same NP number concentration.

Here we report a detailed study on the error of quantification of the concentration of NPs with opposite chirality for the comparative analysis of their interaction with biology. We firstly analyze and evaluate different routes for the determination of NP concentration and consider the error of quantification for each route. Subsequently, we use such error as a threshold to evaluate if the biological variations could be related to merely difference of concentration, or to the chiral properties. The study reveals that only a proper quantification of NPs leads to attributable different biological responses to NPs with equal physicochemical properties except the chiral surface.

## Results

**Synthesis and characterization**. Here we chose CNDs, i.e. quasi-spherical NPs with an amorphous carbon-rich core and diameter under 10 nm, as model system[32]. Our synthetic protocol consists of a microwave-assisted hydrothermal bottom-up synthesis using arginine (Arg) and ethylenediamine (EDA) as precursors[33]. The corresponding nitrogen-doped N-CNDs were shown to possess a nanoscale amorphous core that is covered by an amino-rich surface[33,34]. Additionally, by substituting ethylenediamine with chiral diamines, such as (R,R)- and (S,S)-1,2-cyclohexanediamine, chiral CNDs, termed here R-CNDs and S-CNDs, were

prepared[35]. Electronic circular dichroism of R-CNDs and S-CNDs verified the mirror-image relationship between the two NPs (Supplementary Fig. 1). The basic physicochemical properties of these chiral and achiral CNDs have been demonstrated to be highly similar, such as their diameter as determined by atomic force microscope ($d_{AFM} = 2.47 \pm 0.84$ nm and $2.64 \pm 0.89$ nm for the N-CNDs and R/S-CNDs, respectively)[32,34,35], while their structure and composition, as determined by Fourier-transformed infrared spectroscopy (FT-IR, Supplementary Fig. 2) and X-ray photoelectron spectroscopy (XPS, Supplementary Fig. 3) showed similar multiple oxygen and nitrogen functional groups between them[32,34,35]. However, the absorption and fluorescence emission properties of the three different CNDs (i.e. the achiral N-CNDs and the chiral R/S-CNDs) are different (Supplementary Fig. 4), due to the presence of different surface functionalities and/or emissive traps[32,34,35]. This introduces a general source of error.

**Metric for comparing the different types of CNDs**. As many biological responses to NPs are dependent on the NP dose (i.e. cellular uptake, toxicity, etc.), a metric is needed to apply the N-, S-, and R-CNDs at the same dose to allow for quantitative comparison. Due to the small size and the carbon composition of the CNDs, it is a big challenge to define a reliable metric and thus we will first discuss the different approaches in this regard. In this way, first, the error in not being able to apply the identical amount of N-, S-, and R-CNDs needs to be estimated. Only effects in biological response higher than the error in CND quantification may be considered a significant difference in the biological response to be related to the opposite chirality.

In order to determine number concentrations, i.e. the number of NPs per volume of solution or their molarity (with Avogadro's number being the scaling factor between these two entities), the NPs in a fixed volume of solution need to be counted. For big enough NPs counting can be performed easily with optical microscopy[36]. Due to their small size this however is not possible for the CNDs. In principle, small NPs can be counted by immobilizing them on a surface (optionally with evaporation of the solvent) and by imaging them with high-resolution microscopies, such as atomic force microscopy (AFM) or transmission electron microscopy (TEM). Note that for such single NP imaging, the resolution of the microscope given by the refraction limit does not necessarily need to be better than the size of the NPs. By working with strongly diluted solutions statistically each signal comes from an individual NP and agglomerates can be excluded, and thus counting of NPs can be performed without being able to resolve them. However, as in this case the number of NPs per image is low, there is a huge error in the counting statistics. In the case of the CNDs investigated in this study the relative error in counting, which determines the uncertainty in concentration determination is $\Delta C_{CNDs} C_{CNDs}^{-1} = 43\%$ (Supplementary Table 1). We performed also counting of the CNDs with TEM, which was complicated by their low contrast due to their carbon composition. As TEM with improved refraction limit allows for resolving of individual CND here higher CND concentrations could be used and thus the number of CND counted per image could be increased. However, here agglomeration of the CNDs on the TEM grid occurred, and the relative error in counting, which describes the uncertainty in concentration determination was determined to be $\Delta C_{CNDs} C_{CNDs}^{-1} = 68\%$ (Supplementary Table 2). Another common way for NP counting is nanoparticle tracking analysis (NTA). However, the here used CNDs are below the size limit recommended by the manufacturer of the frequently used Nanosight instrument (the manufacturer Malvern Panalytical recommends NPs > 10 nm diameter) and due

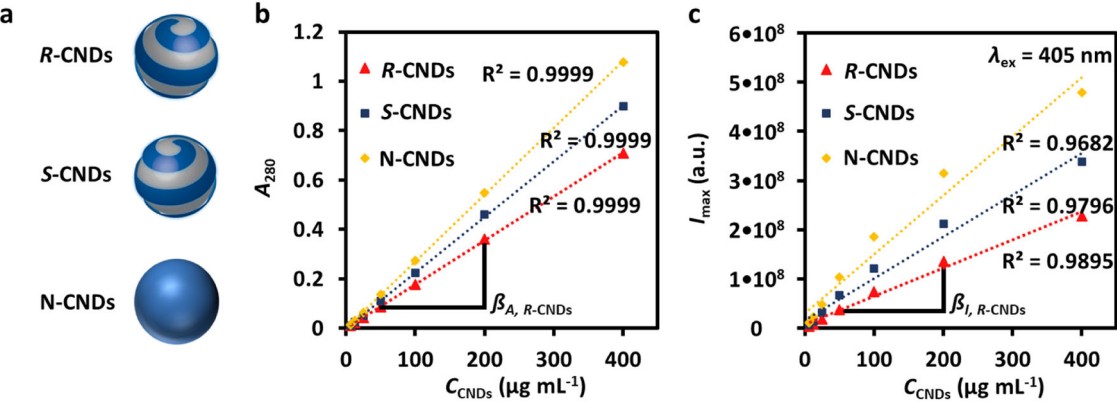

**Fig. 1 Carbon nanodots concentration determination through UV-Vis and fluorescence emission spectrophotometry. a** Sketch of the N-, S-, and R-CNDs. **b** Absorption $A_{280}$ at 280 nm of CNDs dissolved in water at the mass concentration $C_{CNDs}$ (determined by weighting). The $A_{280}(C_{CNDs})$ curve was fitted with linear regression to yield the slope $\beta_{A,j} = \Delta A_{280}(j)\ \Delta C_{CNDs}^{-1}$ (mL μg$^{-1}$) ($j = R$-CND, $S$-CND, N-CND). $R^2$ indicates the fitting reliability with linear regression (perfect fit: $R^2 = 1$). Data for the absorption at 405 nm ($A_{405}$) are shown in Supplementary Fig. 6. **c** Integrated fluorescence emission $I_{max}$ ranging from 425 to 475 nm (excitation wavelength $\lambda_{ex} = 405$ nm) of CNDs dissolved in water at the mass concentration $C_{CNDs}$. The $I_{max}(C_{CNDs})$ curve was fitted with linear regression to yield the slope $\beta_{I,j} = \Delta I_{max}(j)\ \Delta C_{CNDs}^{-1}$ (mL μg$^{-1}$) ($j = R$-CND, $S$-CND, N-CND). From these slopes, first the percentual differences $\Delta\beta_{i,j}$ in the slopes between the R-CND and S-CND sample to the N-CND sample were derived for the absorption and intensity measurements as $\Delta\beta_{i,j} = (\beta_{i,N\text{-}CND} - \beta_{i,j})\ \beta_{i,N\text{-}CND}^{-1}$ ($i = A, I; j = R$-CND, $S$-CND), and then the deviation $\Delta\beta_j$ in these differences between the absorption and intensity measurements were obtained as $\Delta\beta_j = |\Delta\beta_{A,j} - \Delta\beta_{I,j}|$ ($j = R$-CND, $S$-CND). The percentual error in concentration determination was defined as the maximum of these values as $\Delta C_{CNDs}\ C_{CNDs}^{-1} = \max(\Delta\beta_{R\text{-}CND}, \Delta\beta_{S\text{-}CND})$. The values are enlisted in Table 1. The shown data were obtained with batch #1.

to their low fluorescence emission intensity individual CND does not provide sufficient signal to be detected. Only rare agglomerates of CND are detected, leading to artificial huge hydrodynamic diameter (Supplementary Fig. 5). Thus, for the here used CNDs standard NP counting methodologies cannot be applied due to the huge experimental error.

An often-used alternative method to NP counting for the determination of NP concentrations is mass determination. In case of metal NPs the elemental amount of metal from the NPs and thus their concentration can be conveniently determined for example with inductively coupled plasma mass spectrometry (ICP-MS)[37]. However, ICP-MS is not a convenient method for carbon-based NPs such as the here investigated CNDs. Also, simply adjusting the samples to the similar weight of the CNDs is not possible, as apart from experimental errors (limits in the precision of weighting the CND powder; association of water by the hygroscopic CND powder, etc.) the N-, S-, and R-CNDs as prepared in three different syntheses will not have precisely the same mean mass per NP and also the mass distribution of the different samples will not be identical.

For this reason, here concentration determination of the CNDs was performed based on their optical properties, i.e. molar extinction coefficient and quantum yield. Due to different absorption spectra and fluorescence emission intensities, it is however not possible to simply prepare CND samples of similar doses by adjusting the respective concentrations to yield solutions with the same absorption or fluorescence intensity. In order to estimate the error in concentration determination, we plotted the absorption at 280 nm ($A_{280}$) and the integrated fluorescence emission intensity $I_{max}$ from 425 to 475 nm upon excitation at 405 nm of the CND samples of three different batches at different mass concentrations $C_{CNDs}$, which were determined by dissolving CND powder of know mass in a known volume of Milli-Q water (Fig. 1). Note that these measured absorptions and emissions are not entirely linear[38], which introduces an additional complication. The maximum difference in concentrations for the same sample, determined with the two methods, was considered as an error in concentration determination $\Delta C_{CNDs}\ C_{CNDs}^{-1} = 18\%$ (Table 1). In total, five different batches were analyzed and the mean error in concentration determination was found to be

$\Delta C_{CNDs}\ C_{CNDs}^{-1} = 22\%$ (Supplementary Table 3). This error is clearly better than the errors obtained by AFM and TEM counting and thus, in our hands, the best way to determine the CND concentrations. Consequently, only biological effects bigger than 22% of the different CND samples will be considered to be significantly above the error in concentration determination.

To probe the interaction of the CNDs with proteins and cells, the CND samples were adjusted to have the same absorption at 280 nm. As the R-, S-, and N-CNDs at the same concentration $C_{CNDs}$ as determined by weighting have slightly different absorption (Fig. 1b), we diluted the two samples with higher absorption than the third sample, until the R-, S-, and N-CNDs samples had the same absorption intensity at 280 nm. Upon dilution, the mass concentration of the CND samples has been slightly changed. We thus refer to the concentrations of the CNDs in the following as adjusted concentrations $C'_{CNDs}$, which refers to their absorption value. For the undiluted sample $C'_{CNDs} = C_{CNDs}$, for the samples diluted to match the absorption, the adjusted concentration $C'_{CNDs}$ is defined as equal to $C'_{CNDs}$ of the undiluted sample. For details, we refer to Supplementary Fig. 7.

For the uptake studies, the amount of internalized CNDs was quantified by their fluorescence. However, as the different types of CNDs have different quantum yields, from the absolute detected fluorescence intensity $I$ as detected for the different CND samples, the CND concentration $C'_{CND}$ (which is proportional to their absorption) could not be directly derived. Thus, a correction factor $X$ had to be applied (Supplementary Table 4, Supplementary Figs. 8 and 9), that the different CND samples had the same fluorescence $I' = X \times I$ at the same concentration $C'_{CND}$. Note that the correction factor $X$ had to be determined in the cell culture medium and for the used devices with which fluorescence was detected (i.e. fluorescence spectrometer, flow cytometry, confocal microscopy). Also, background correction was applied. Only by using this double correction ($C_{CND} \rightarrow C'_{CND}$ and $I \rightarrow I'$) the uptake of CND by cells could be compared for the different types of CNDs.

**Role of the protein corona.** As first impact of the interaction of the CNDs with biology, we chose the adsorption of different proteins. By measuring protein concentration-dependent size

**Table 1 Error analysis in concentration determination by absorption ($A_{280}(C_{CNDs})$) and fluorescence intensity ($I_{max}(C_{CNDs})$) measurements.**

| $\beta_{A,R-CND}$ (mL µg⁻¹) | $\beta_{A,S-CND}$ (mL µg⁻¹) | $\beta_{A,N-CND}$ (mL µg⁻¹) | $\beta_{I,R-CND}$ (mL µg⁻¹) | $\beta_{I,S-CND}$ (mL µg⁻¹) | $\beta_{I,N-CND}$ (mL µg⁻¹) | $\Delta\beta_{A,R-CND}$ | $\Delta\beta_{A,S-CND}$ | $\Delta\beta_{I,R-CND}$ | $\Delta\beta_{I,S-CND}$ | $\Delta\beta_{R-CND}$ | $\Delta\beta_{S-CND}$ | $\Delta C_{CNDs} C_{CNDs}^{-1}$ |
|---|---|---|---|---|---|---|---|---|---|---|---|---|
| 0.00177 | 0.00225 | 0.00270 | 569422 | 842279 | 1187647 | 0.34 | 0.17 | 0.52 | 0.29 | 0.18 | 0.13 | 0.18 |

The parameters are defined in Fig. 1. The shown data were obtained with batch #1.

increase of the CND−protein conjugates by fluorescence correlation spectroscopy[39–41], absolute quantifiers describing the interaction could be obtained, namely the apparent dissociation constant $K_D$, the maximum number of proteins $N_{max}$ that can bind per CND based on the Hill model, and the maximum size increase in hydrodynamic radius $\Delta r_{h,max}$ upon protein adsorption. $K_D$ describes the protein concentrations at which half of the CND surface is covered with proteins. As model proteins, three representative serum proteins, i.e. human serum albumin (HSA), alpha microglobulin (α2M), and transferrin (Tf) were used. Data of batch #1 are shown in Fig. 2, and the data of the other two batches are depicted in Supplementary Fig. 10.

The data shown in Fig. 2 indicate that proteins in general only weakly bind to the CNDs. The $K_D$ value gives the protein concentration, which is needed to half-saturate the CND surface (i.e. to have half of the maximum possible number of bound proteins). In comparison to other NPs the $K_D$ values are higher, meaning that the CNDs are worse binders for the proteins than the other NPs. In the case of HSA $K_D$ values of around 5.1 µM have been obtained for polymer-coated FePt NPs[39,42], which is lower than the $\langle K_D \rangle$ of 32.3 µM (Table 2) of the CNDs. Transferrin binds only weakly to the CNDs and under the explorable concentration range no saturation could be achieved, meaning that $\langle K_D \rangle > 1000$ µM. This is much higher than $K_D$ values of around 26 µM[40,42], which have been obtained for polymer-coated FePt NPs. Alpha microglobulin does not adsorb to any of the tested CND surfaces within the tested concentration range. Thus, as with Tf, no quantitative values could be detected for the $K_D$ value, which will be >1000 µM. Only HSA resulted to adsorb sufficiently well to all CND surfaces and only for this protein a quantitative analysis based on $K_D$ was possible. The derived parameters for HSA for N-, S-, and R-CNDs of the three different batches are enlisted in Supplementary Table 5. A possible interference for quantitative analysis of the binding of proteins to NP surfaces is NP agglomeration[43–45]. In case the size increase of the NPs upon adsorption of protein is much bigger than the size of the proteins (note that in many cases a monolayer adsorption of proteins covering the NP surface has been described), then this may be due to agglomeration effects. As shown in Table 2 (and Supplementary Table 5), the maximum change in hydrodynamic CND radius upon saturation of the CND surface with HSA ($\Delta r_{h,max}$) is around 2.5 nm (Supplementary Table 5), which corresponds to the size of one HSA molecule, in good agreement with previous studies[39,46]. Of note, the size of one HSA molecule is much bigger than the size of one CND with a hydrodynamic radius of $r_{h,0}$ of around 0.7–1.0 nm (Supplementary Table 5). Due to the small size of the CNDs one CND can adsorb only 1–2 HSA molecules ($N_{max}$, Table 2 and Supplementary Table 5). We note that there is variation in $K_D$ values between the different CND batches (Supplementary Fig. 10 and Supplementary Table 5). In fact, for batch #2 and batch #3 under the explorable HSA concentration range, no saturation of the CND surface with HSA could be reached, i.e. the $r_h(c_{HSA})$ curves did not reach saturation. Under these conditions, the determination of $K_D$ values is extremely prone to errors (Supplementary Table 5), and thus we only considered the values of batch #1 (Table 2) for quantitative analysis. However, the

batch-to-batch variability has to be taken into account when discussing the biological significance of the results. The data reported in Table 2 demonstrate that S-CNDs are significantly worse binders to HSA than R-CNDs (batch #1: $K_D$ of 39.9 µM versus 22.7 µM). The difference between both is $\Delta K_D\langle K_D \rangle^{-1} = 0.36$, which is bigger than the error associated with determining the CND concentration $\Delta C_{CNDs} C_{CNDs}^{-1} = 22\%$ (Supplementary Table 3), though the CND concentration does at any rate only moderately influence the results for the $K_D$ determination[41]. As mentioned, as no saturation of the CNDs of batch #2 and batch #3 with HSA could be achieved these $K_D$ values are unreliable. Still, also for batch #2 and batch #3 the $K_D$ value for R-CNDs is lower than for S-CNDs. The HSA data for the N-CNDs are in the middle for batch #1. However, for batch #2 and batch #3 the $K_D$ values seem to be much higher than for the R-CNDs and S-CNDs in these batches (Supplementary Table 5). Again, as saturation of the CNDs with HSA is not though out of three CND batches only one could be quantitatively evaluated. Of note, for these batches (#2 and #3) also the hydrodynamic radii $r_{h,0}$ of the N-CNDs are higher than in the other cases (Supplementary Table 5), which might be a reason for the different $K_D$ values. Also in the UV-Vis absorption spectra of batch #4 of the N-CNDs some agglomeration (e.g. scattering at high wavelength) is visible (Supplementary Fig. 11). Thus, protein adsorption on N-CNDs might be influenced by slight agglomeration and differences to the R- and S-CNDs cannot be unequivocally related to changes in chirality.

We thus can summarize that CNDs in comparison to other NPs are very weak binders to proteins, and from the three investigated proteins here only HSA formed a clear protein corona. For HSA there is a significantly better binding of R-CNDs to the CNDs than of S-CNDs. The HSA binding of N-CNDs had large batch−batch variations and there was the same agglomeration, and thus these data cannot be interpreted quantitatively.

We want to mention that our data refer only to three selected serum proteins. As with our method, i.e. FCS, we only detect changes in the hydrodynamic diameter upon protein adsorption, upon exposure to blood we would not be able to tell which proteins had adsorbed and caused the increase in size of the NPs. In order to detail the composition of the protein corona typically mass spectroscopy analysis is performed[47]. However, for such measurements first unbound excess proteins have to be removed. For the small CNDs as investigated here to which only few proteins can weakly bind, such purification may significantly change the protein composition left on the NP surface[41]. In contrast, diffusion measurements with FCS are performed in situ, without the need for purification. While such measurements do not allow for telling the composition of the adsorbed protein corona, and thus are best carried out in different solutions containing only one type of model protein, diffusion measurements can still verify protein corona formation in blood[48].

**Cellular uptake.** Cellular uptake of the different types of CNDs (here batches #3, #4, and #5 were used) was quantified with previously established methods[49–51] with two cell lines (see "Methods"). HeLa cells were used as a standard model system,

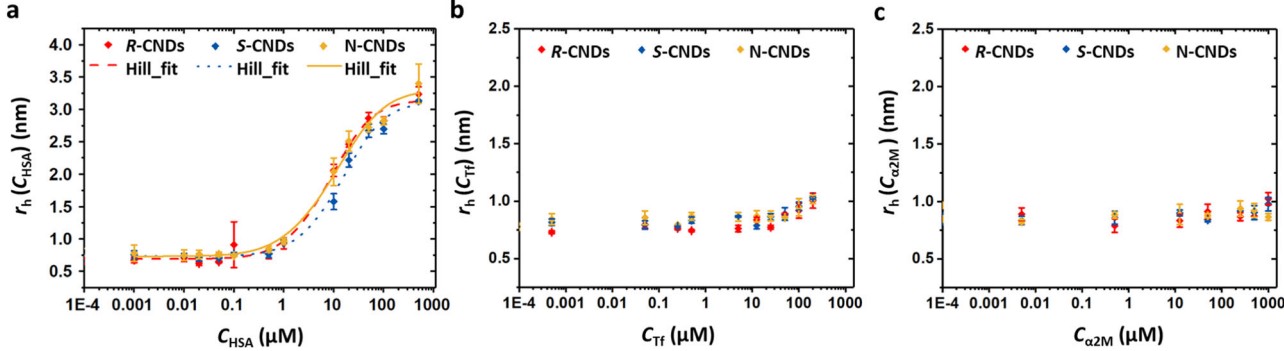

**Fig. 2 Change of hydrodynamic radius $r_h$ of CNDs (batch #1) as recorded in phosphate-buffered saline (PBS) in dependence of the protein concentration. a** HSA; **b** Tf; **c** α2M. From the plots the fit parameters $K_D$, $N_{max}$, $n$, $r_{h,0}$, and $\Delta r_{h,max}$ were obtained, which are listed in Table 2. $K_d$ is the apparent dissociation constant of the CND-protein complex, $N_{max}$ is the maximum number of bound proteins per CND under saturation conditions, $n$ is the Hill coefficient, $r_{h,0}$ is the hydrodynamic radius of the CNDs without attached proteins, and $\Delta r_{h,max}$ is the difference in effective hydrodynamic radius between CNDs saturated with proteins and CNDs without attached proteins. The shown data were obtained with batch #1. Results are shown as mean values with error bars (i.e. the corresponding standard deviations) from three independent samples ($n = 3$) over three independent experiments.

**Table 2 Parameters describing the interaction of CNDs and HSA.**

| CND type | $K_D$ (μM) | $N_{max}$ | $n$ | $r_{h,0}$ (nm) | $\Delta r_{h,max}$ (nm) | $\langle K_D \rangle$ (μM) | $\Delta K_D \langle K_D \rangle^{-1}$ |
|---|---|---|---|---|---|---|---|
| *R*-CNDs | 22.7 ± 5.3 | 1.2 ± 0.3 | 1.2 ± 0.2 | 0.69 ± 0.05 | 2.4 ± 0.1 | 32.3 | −0.30 |
| *S*-CNDs | 39.9 ± 9.5 | 1.2 ± 0.3 | 1.2 ± 0.2 | 0.73 ± 0.04 | 2.3 ± 0.1 | | +0.24 |
| *N*-CNDs | 34.3 ± 10.4 | 2 ± 0.6 | 1.0 ± 0.1 | 0.73 ± 0.05 | 2.5 ± 0.1 | | +0.06 |

Apparent dissociation constant $K_D$, maximum number $N_{max}$ of HSA molecules adsorbed per CND, cooperativity parameter $n$, hydrodynamic radius $r_{h,0}$ of the CNDs without exposure to HSA, and maximum increase of hydrodynamic radius $\Delta r_{h,max}$ of the CNDs upon saturation with HSA. The error is the standard error as obtained by fitting of the data with the Hill-Model. $\Delta K_D$(*S*-CNDs) $\langle K_D \rangle^{-1} - \Delta K_D$(*R*-CNDs) $\langle K_D \rangle^{-1} = 0.24 - (−0.30) = 0.54$. Data were recorded with batch #1.

which is widely spread in general uptake studies[52]. As second system THP-1 macrophages as derived by the differentiation of the human monocytic leukemia cell line THP-1[53] were used, modeling an exposure scenario which CNDs could encounter in vivo. Viability assays were performed[50], which demonstrated that under the used incubation conditions the CNDs are biocompatible (see "Methods").

Uptake of CNDs by cells as quantified by the mean CND fluorescence $I'$ per cell was detected with two independent methods, flow cytometry (see "Methods") and confocal microscopy (see "Methods"), as there may be discrepancies between the results of these two detection techniques[54]. Also, time- and concentration-dependence of CND uptake was quantified[49], as well in serum-free as in serum-supplemented media. Probing such a range of different parameters serves as an internal control that CNDs were uptaken like other NPs. CNDs showed the expected general uptake behavior[49]: uptake of CNDs increased with increasing incubation times, with increasing CND concentration, and was higher in serum-free than in serum-supplemented medium (see "Methods"). In this way, differences in the uptake of the different types of CNDs are due to differences in the CNDs and not due to other factors. The main findings of the uptake quantification study are summarized in Fig. 3. Here the time-dependence of the uptake is shown. The corresponding concentration-dependence is presented in the "Methods" section. The data show that consistently there is (at the same incubation concentration $C'_{CND}$) lowest internalization of N-CNDs, and highest internalization for S-CNDs, as well for HeLa cells as for THP-1 derived macrophages (Fig. 3a versus Fig. 3b), quantified by both flow cytometry and confocal microscopy (Fig. 3b versus Fig. 3d). The question is now whether these differences in-between the different types of CNDs are significant. For this, the values at $t = 48$ h are analyzed in Table 3.

For N-CNDs there was always lower internalization than for S-CNDs and R-CNDs. As there was some agglomeration for the N-CNDs as discussed above this effect most likely is not related to the chirality of the CNDs, but to their colloidal stability, and thus will not be discussed further. For all investigated conditions there was more uptake for S-CNDs than for R-CNDs. Note, that the data shown in Table 3 are not "cherry-picked", and in fact, under different CND concentrations the uptake difference between R-CNDs and S-CNDs was even higher (Supplementary Fig. 13). This is an important internal control, that results are not related to only one particular incubation condition but are valid in general. While the data shown in Fig. 3 and Table 3 show that S-CNDs internalize best, for making a final statement the differences in uptake need to be related to the experimental errors. For example, while for THP-1 derived macrophages the tendency as obtained from flow cytometry is the same as from confocal microscopy data, the difference between both types of CNDs is more pronounced for the confocal microscopy data. In Fig. 1 we pointed out that there is an uncertainty of CND concentration determination of $\Delta C'_{CNDs}$ $C'^{-1}_{CNDs} = 22\%$. From Table 3 the difference in uptake between S-CNDs and R-CNDs for HeLa cells is $\Delta I' \langle I' \rangle^{-1} = 16\%$ and 12% for serum-supplemented and serum-free incubation conditions, respectively. According to our analysis the change in uptake difference for HeLa cells might be caused by uncertainty in concentration determination ($\Delta I' \langle I' \rangle^{-1} < \Delta C'_{CNDs}$ $C'^{-1}_{CNDs}$) and thus no difference in the uptake behavior based on chirality should be claimed. In contrast, from Table 3 the difference in uptake between S-CNDs and R-CNDs for THP-1 derived macrophages is $\Delta I' \langle I' \rangle^{-1} = 24\%$ and 38% for serum-supplemented and serum-free incubation conditions (flow cytometry data) and $\Delta I' \langle I' \rangle^{-1} = 88\%$ and 110% for serum-supplemented and serum-free incubation conditions (confocal microscopy data), respectively. Thus, these differences are clearly above the error of concentration determination ($\Delta I' \langle I' \rangle^{-1} > \Delta C'_{CNDs}$ $C'^{-1}_{CNDs}$) and are thus related to the different chirality of the CNDs. As macrophages are designed to interact with "intruding" molecules/particles/materials, it is plausible that they can better distinguish "what to

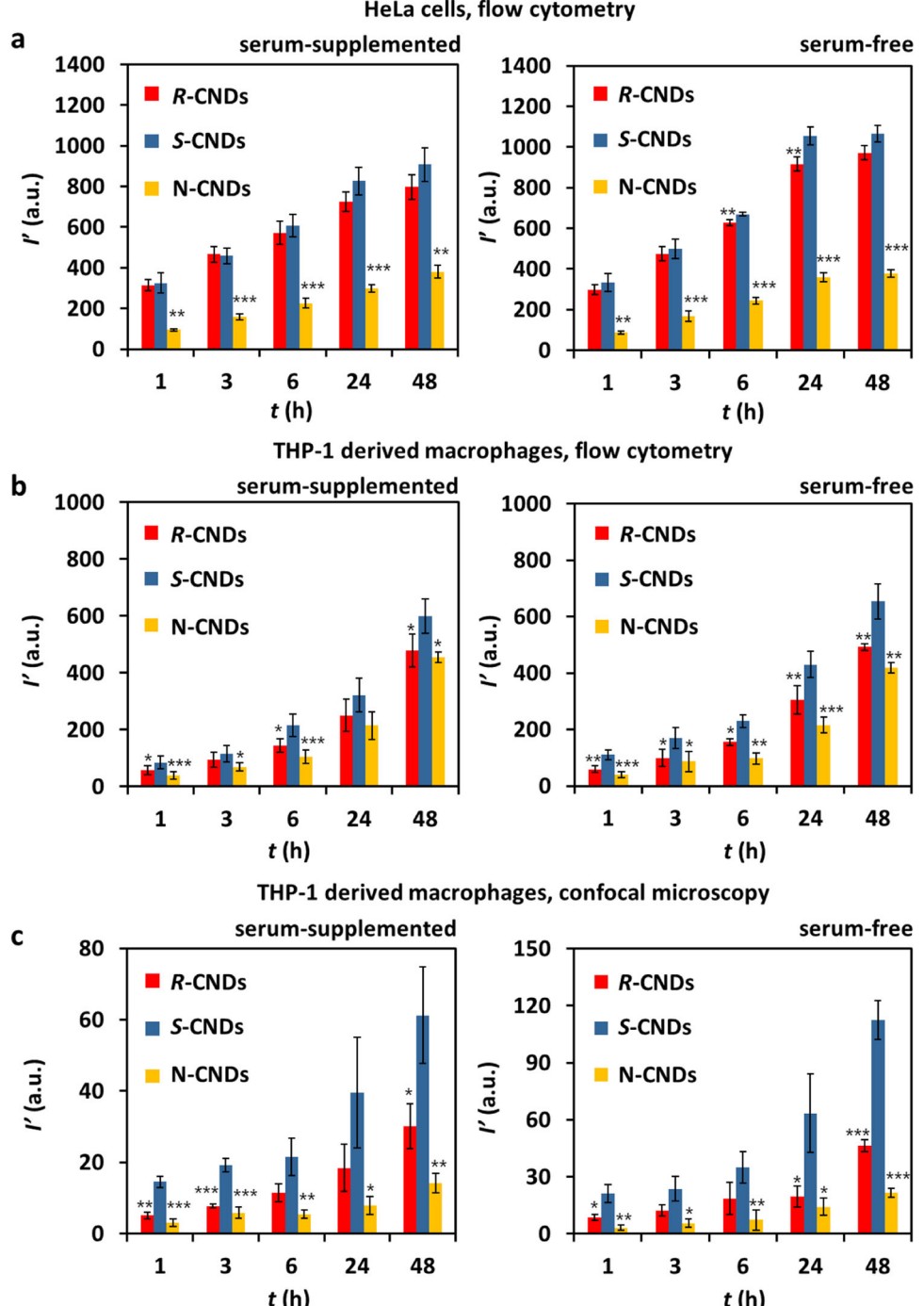

**Fig. 3 Uptake of CNDs as quantified by the mean CND fluorescence $I'$ per cell.** Cells were incubated for different times $t$ at a CND concentration of $C'_{CNDs} = 400\ \mu g\ mL^{-1}$ in serum-supplemented (10% FBS) and serum-free medium. **a** HeLa cells, $I'$ detected by flow cytometry, see also Supplementary Fig. 22. **b** THP-1 derived macrophages, $I'$ detected by flow cytometry, see also Supplementary Fig. 24. **c** THP-1 derived macrophages, $I'$ detected by confocal microscopy, see also Supplementary Fig. 12. Results are shown as mean value ± standard deviation (s.d.) from three independent samples ($n = 3$) over three independent experiments. $P$ values were analyzed by Student's $t$ test with two-tailed distribution and two-sample equal variance. $*P < 0.05$, $**P < 0.01$, $***P < 0.001$.

take up", i.e. better differential of surface differences between $S$-CNDs and $R$-CNDs. We thus can summarize that for THP-1 derived macrophages there is significantly higher uptake from $S$-CNDs than for $R$-CNDs.

Apart from the uptake quantification we also performed colocalization assays and chemical blocker for known pathways of endocytosis (see "Methods"). All types of internalized CNDs colocalized to a high degree with lysosomes (Supplementary Fig. 15). Cellular uptake of the CNDs was blocked at 4 °C, suggesting that the internalization of the CNDs is an energy-dependent process (see "Methods")[55]. The three types of CNDs were taken up by both different cell types largely via a

**Table 3 Mean fluorescence intensity $I'$ detected per cell after $t = 48$ h incubation at $C'_{CNDs} = 400\ \mu g\ mL^{-1}$, corresponding to the last time point in Fig. 3.**

| | Serum-supplemented medium | | | Serum-free medium | | |
| --- | --- | --- | --- | --- | --- | --- |
| | HeLa | THP-1 | | HeLa | THP-1 | |
| | Flow cytometry | Flow cytometry | Confocal microscopy | Flow cytometry | Flow cytometry | Confocal microscopy |
| $I'$(R-CNDs) (a.u.) | 798 | 480 | 30.2 | 971 | 492 | 46.4 |
| $I'$(S-CNDs) (a.u.) | 907 | 601 | 61.3 | 1065 | 654 | 112.5 |
| $I'$(N-CNDs) (a.u.) | 381 | 455 | 14.2 | 378 | 149 | 21.5 |
| $\langle I' \rangle$ | 695 | 512 | 35.2 | 805 | 432 | 60.1 |
| $\Delta I'$(R-CNDs) $\langle I' \rangle^{-1}$ | 0.15 | −0.06 | −0.14 | 0.21 | 0.14 | −0.23 |
| $\Delta I'$(S-CNDs) $\langle I' \rangle^{-1}$ | 0.30 | 0.17 | 0.74 | 0.32 | 0.52 | 0.87 |
| $\Delta I'$(N-CNDs) $\langle I' \rangle^{-1}$ | −0.45 | −0.11 | −0.60 | −0.53 | −0.65 | −0.64 |
| $\Delta I'$(S-CNDs) $\langle I' \rangle^{-1}$ – $\Delta I'$(R-CNDs) $\langle I' \rangle^{-1}$ | 0.16 | 0.24 | 0.88 | 0.12 | 0.38 | 1.10 |

The full tables are provided in the Supplementary Information (Supplementary Tables 6–8). For all three types of CNDs the mean value for each incubation condition is given as $\langle I' \rangle = (I'(R\text{-CNDs}) + I'(S\text{-CNDs}) + I'(N\text{-CNDs}))\ 3^{-1}$. Deviations are calculated as $\Delta I'(j) = (I'(j) - \langle I' \rangle)\langle I' \rangle^{-1}$ ($j = $ R-CND, S-CND, N-CND). $\Delta I' \langle I' \rangle^{-1} = \Delta I'(S\text{-CNDs})\ \langle I' \rangle^{-1} - \Delta I'(R\text{-CNDs})\ \langle I' \rangle^{-1}$ tells the relative difference in fluorescence intensity per cell between S-CNDs and R-CNDs.

phagocytosis pathway, while N-CNDs could also be endocytosed via a clathrin-associated endocytosis pathway, similar to the results obtained for the THP-1 derived macrophages (see "Methods"). The different cellular uptake behavior of the N-CNDs may be related to their lower colloidal stability.

## Discussion

Correlating the interaction of NPs with cells with the physicochemical properties of the NPs is by far not trivial. Several physicochemical properties may be entangled[56], and many time effects may not be due to a primary physicochemical parameter (such as e.g. chirality), but due to colloidal stability (as here in the case of the N-CNDs). In fact, we have shown here by FCS measurements that for the R- and S-CNDs chirality does not affect colloidal stability and only because for this case entanglement of chirality and colloidal stability was ruled out differences in biological effects can be related to chirality as physicochemical parameter. For the N-CNDs there was an effect on colloidal stability and thus differences in their biological effects cannot be related to their non-chiral nature.

For many types of NPs concentration determination is not unequivocal[31], and thus errors in the metric (here $\Delta C'_{CNDs}\ C'^{-1}_{CNDs}$) need to be taken as threshold whether detected differences in the interaction of the NPs with cells can be related to certain physicochemical parameters. In the present case, the uptake of S-CNDs is significantly higher than that of R-CNDs for THP-1 derived macrophages, and differences in uptake can be related to the different chirality between both types of CNDs. For the case of HeLa cells, the same tendency was observed, but differences in uptake might be also due to uncertainties in concentration determination.

Taking into account the protein corona data, HSA binds significantly better to R-CNDs than to S-CNDs (Table 2: $K_D$(R-CNDs) $= 22.7\ \mu M < K_D$(S-CNDs) $= 39.9\ \mu M$). It is not directly obvious why the different chirality between R-CNDs and S-CNDs leads to different adsorption behavior of HSA, as concerning the chemical composition, charge, etc. but surfaces are identical. As the surface of the CNDs will not be smooth and homogenous, most likely the different chirality has some local effect on the arrangement of the organic shell, like a different local density of the functional groups or different local conformation, which would explain why HSA binds differently to both surfaces.

R-CNDs are endocytosed to a significantly lesser extent by THP-1 derived macrophages than S-CNDs (Table 3: $I'$(R-CNDs) $< I'$(S-CNDs)). An easy way to interpret this scenario could be the following. There is more protein corona for R-CNDs.

The presence of a protein corona for many NPs is often associated with lower uptake by cells (in fact there is lower uptake in serum-supplemented conditions where a protein corona can be formed than in serum-free conditions), which would explain the reduced uptake for the R-CNDs. However, this interpretation might be too simple. In comparison to other NPs the CNDs are very poor binders to proteins, which will be influenced by their small size. Only for HSA, but not for Tf and α2M saturation in binding could be achieved (Fig. 2). However, also for HSA there is on average only one single HSA molecule associated with each CNDs (Table 2; $N_{max}$ parameter). Differences in uptake ($\Delta I'$(S-CNDs) $\langle I' \rangle^{-1}$ versus $\Delta I'$(R-CNDs) $\langle I' \rangle^{-1}$) were larger in serum-free than in serum-supplemented medium. While also under serum-free culture conditions presence of some proteins in the culture medium cannot be excluded, it is unlikely that at low protein concentrations ($c_p$ of proteins in serum-free medium $\ll K_D$) a difference in the attachment of proteins from the medium between S-CNDs and R-CNDs is the reason for their different uptake by cells, in particular as the CNDs are poor binders for proteins. The surface of cells is not homogeneous, and the lipid bilayer is patterned with proteins and sugars. Before a NP is internalized by a cell, it first needs to bind to the cell surface, where it may dwell for some time until it is endocytosed[57]. We thus speculate that in addition to protein corona-related effects there might be better (non-specific) adsorption of R-CNDs to the cell surface than of S-CNDs, which then would relate to higher uptake. This demonstrates that while the general picture of particle uptake is quite well established[58], it is not that all details would be understood.

The data shown here demonstrate that also under most stringent considerations of errors in the concentrations determination of CNDs, it has been shown that chirality may affect the protein corona formation and in vitro cellular uptake of CNDs to an extent of >20%. It thus can be speculated that this difference would also influence the in vivo interaction of CNDs. Chirality itself does not influence the most important physicochemical properties of CNDs, such as fluorescence and colloidal properties. By using CNDs of different chirality thus different biodistributions of otherwise identical CNDs might be obtained.

## Methods

### Characterization and concentration determination of the CNDs

*Basic optical characterization of the CNDs.* Different carbon nanodots (CNDs) were synthesized according to previously published protocols, i.e. N-CNDs[34], and R-CNDs and S-CNDs[35]. Firstly, the optical properties of the CNDs were

characterized. Briefly, the obtained lyophilized solids of CNDs were first weighted and then fully dissolved in filtered Milli-Q water to form stock solutions with a concentration of $C_{CNDs} = 10\,mg\,mL^{-1}$. Then, the stock solutions were further diluted to $C_{CNDs} = 100\,\mu g\,mL^{-1}$ with filtered sterilized water for absorbance and fluorescence spectra measurements (Supplementary Fig. 4) using an UV-Vis absorption spectrophotometer (Agilent 8453, Agilent technologies, Australia) and a fluorescence spectrometer (Fluorolog-3, Horiba Jobin Yvon, USA). In the absorption spectra the R-CNDs and S-CNDs displayed two absorption peaks, while the N-CNDs only had one peak, at the same position as the R-CNDs and S-CNDs at 280 nm. Concerning the fluorescence properties, the three CNDs demonstrated a slight excitation-dependent emission shift upon excitation at different wavelength $\lambda_{ex}$ ranging from 330 to 440 nm.

*Concentration-dependent absorption/fluorescence intensity measurements.* In order to account for different absorption and emission properties of the different CNDs, the concentration-dependent adsorption and fluorescence spectra were measured for the different CND samples. For this objective, the CND solutions with concentrations ranging from $C_{CNDs} = 0.78\,\mu g\,mL^{-1}$ to $400\,\mu g\,mL^{-1}$ were prepared by diluting the CND stock solutions with filtered Milli-Q water. Then, UV-Vis absorption spectra $A(\lambda)$ and fluorescence spectra $I(\lambda)$ ($\lambda_{ex} = 405\,nm$) were recorded (Supplementary Fig. 26). The three different types of CNDs all possessed a dose-dependent absorbance and fluorescence behavior as expected, i.e. the adsorption and fluorescence linearly decreased with more diluted CNDs solutions. The absorption values $A_{280} = A(\lambda = 280\,nm)$ plotted against the CND concentration are shown in Fig. 1 in the main manuscript. In order to probe the influence of the wavelength at which the absorption is measured for the error analysis, the same evaluation as performed for Fig. 1b in the main paper was also performed for the absorption $A_{405}$ at 405 nm of the CNDs (Supplementary Fig. 6). As $A_{405} \ll A_{280}$ all further evaluation was performed with $A_{280}$.

*Concentration adjustment and fluorescence intensity correction.* As the CNDs at different mass concentration $C_{CNDs}$ (as determined by weighting) showed different absorption (Supplementary Fig. 6), the concentrations were adjusted to lead to the same absorption at 280 nm. This was done to allow further concentration determination via absorption measurements. For this, the adsorption values of the CND solutions were determined at 280 nm and the CND solutions with higher adsorption values (which was always the R-CNDs) were diluted with filtered Milli-Q water until achieving all samples had the same absorption $A_{280}$. The R-CND solution remained undiluted at concentration $C_{CNDs}$. In the following the adjusted concentrations of the S-CND and N-CND solutions were assumed to have the same concentration as the R-CND solution. Here the "same concentration" refers to equal absorption at 280 nm. These absorption-based concentrations are referred to as adjusted concentrations $C'_{CNDs}$. $C'_{CNDs}(S\text{-}CNDs) = C'_{CNDs}(N\text{-}CNDs) = C'_{CNDs}(R\text{-}CND) = C_{CNDs}(R\text{-}CND)$. An example is shown in Supplementary Fig. 7. In order to determine the error in concentration determination absorption $A(\lambda)$ and emission spectra $I(\lambda)$ were recorded in dependence of the adjusted concentrations $C_{CNDs}$. The spectra are displayed for the five different batches of CNDs used in this study in Supplementary Figs. 18 and 19. The concentration-dependence of the absorption at 280 nm and of the fluorescence emission as derived from the spectra is plotted for all five batches in Supplementary Fig. 27. Based on Supplementary Fig. 23 error analysis was performed as described in Fig. 1 of the main manuscript, and the results are displayed in Supplementary Table 3. As can be seen in Supplementary Figs. 7 and 27 at the same adjusted concentration $C'_{CND}$ the R-CND, S-CND, and N-CND samples have different fluorescence emission intensities. This needs to be considered when quantifying the uptake of CNDs by their fluorescence with flow cytometry and confocal microscopy. For flow cytometry was collected with a 450 nm/50 nm bandpass filter, and for confocal microscopy with an LP 420 nm long pass filter (Supplementary Fig. 7). In this way correction factors $X$ taking into account the different fluorescence intensities at the same adjusted concentrations are defined. $X_{S/R} = \langle I(S\text{-}CNDs)\rangle \langle I(R\text{-}CNDs)\rangle^{-1}$ is the ratio of the integrated S-CND fluorescence and the integrated R-CND fluorescence, at the same adjusted concentration of S-CNDs and R-CNDs. The integration range was used emulating the flow cytometry and confocal microscopy filters. $X_{S/N} = \langle I(S\text{-}CNDs)\rangle \langle I(N\text{-}CNDs)\rangle^{-1}$ is the ratio of the integrated S-CND fluorescence and the integrated N-CND fluorescence, at the same adjusted concentration of S-CNDs and N-CNDs. The resulting values are enlisted for all five used batches in Supplementary Table 4.

*Concentration determination by CND counting with atomic force microscopy (AFM).* Diluted CND solutions of concentration $C_{CNDs}$ of R-CNDs and S-CNDs were drop-casted on mica substrates for AFM analysis. Images were acquired by tapping mode AFM (Nanoscope IIIa, VEECO Instruments) on a surface area $A_{scan}$ of 25 $\mu m^2$ (Supplementary Figs. 28 and 29). The number $N_{CNDs}$ of CNDs as identified in the image (by looking at the height profiles) was counted for samples prepared by two concentrations per typology of CNDs (R- and S-) and is displayed in Supplementary Table 1. The respective mean values $\langle N_{CNDs}\rangle$ and standard deviations $\Delta N_{CNDs}$ were then calculated. Taking together all determined four values for $\Delta N_{CNDs}\langle N_{CNDs}\rangle^{-1}$ leads to a mean value of $\Delta N_{CNDs}\langle N_{CNDs}\rangle^{-1} \approx 0.21$. The normalized error in NP counting corresponds to the normalized error in concentration determination $\Delta c_{CND} c_{CND}^{-1}$. Note that the real error even might be

higher. This point is better rationalized if we consider the mass per CND to compare the theoretical and calculated number of CNDs per substrate area. Considering the similar average size of CNDs we calculated the average NP mass of CNDs considering their sphericity and the density of amorphous carbon ($\rho_C = 3.50\,g\,cm^{-3}$)[35]. By calculating the expected and measured number of particles on substrate, $\langle n_{CNDs}\rangle = \langle N_{CNDs}\rangle A^{-1}$, is thus possible to make a direct comparison of the two quantities. As reported in Supplementary Table 1, the calculated and expected $\langle n_{CNDs}\rangle$ values are different by several order of magnitude and this result may be influenced by the drop casting deposition process[59] that could be cause of: (i) inhomogeneous distribution of NPs on the surface, (ii) formation of aggregates during the solvent evaporation. The following paragraph, treating the CNDs quantification trough transmission electron microscopy (TEM), further remarks the difficulty on calculating these NPs trough microscopy techniques using an analogous concept.

*Concentration determination by CND counting with transmission electron microscopy (TEM).* Transmission electron microscopy (TEM) measurements were performed using a Jeol JEM-1011 instrument operating at 100 kV. 2 μL of the according CND solution ($C_{CNDs}(R\text{-}CNDs) = 2.0\,mg\,mL^{-1}$; $C_{CNDs}(S\text{-}CNDs) = 2.8\,mg\,mL^{-1}$) were drop-casted onto a copper grid (400 mesh, diameter 3.05 mm) coated with amorphous carbon. As can be observed from the TEM-micrographs shown in Supplementary Figs. 30 and 31, homogeneous coating was not achieved. For the R-CNDs different aggregates were observed, whereas for the S-CNDs some areas with dispersed CNDs were also found. It is not known, how much of the aggregates form during drying on the TEM grid. On both samples it was possible to differentiate single CNDs; however, due to the limited contrast, an exact determination of CND size was not possible. Based on micrograph analysis with ImageJ we obtained $d_{TEM} \approx 1.5 \pm 0.4\,nm$ for R-CNDs ($N = 216$ CNDs investigated) and $d_{TEM} \approx 2.4 \pm 0.9\,nm$ for S-CNDs ($N = 255$ CNDs investigated). These values are compatible with those obtained by AFM ($d_{AFM}$) and discussed in the main text. We emphasize, however, that due to the limited number of observed CNDs and limited contrast, the diameters as determined with TEM should be considered a rough estimate. We used the determined CND diameters to calculate theoretical concentrations, assuming sphericity. With the density of amorphous carbon $\rho_C = 3.50\,g\,cm^{-3}$ (diamond has a similar density of $\rho_C = 3.51\,g\,cm^{-3}$) and the weight concentrations ($C_{CNDs}(R\text{-}CNDs) = 2.0\,mg\,mL^{-1}$; $C_{CNDs}(S\text{-}CNDs) = 2.8\,mg\,mL^{-1}$) we obtain a molar concentration of $c_{CNDs} \approx 540\,\mu M$ for R-CNDs and $c_{CNDs} \approx 180\,\mu M$ for S-CNDs. Because of the limited accuracy of CND diameter determination, also these concentrations must be considered as a rough estimate. In the TEM micrographs of S-CNDs we find $\langle n_{CNDs}\rangle = \langle N_{CNDs}\rangle A_{scan}^{-1} \approx 3014$ CNDs $\mu m^{-2}$ on average (Supplementary Table 2). For R-CNDs we find $\langle n_{CNDs}\rangle \approx 1920$ CNDs $\mu m^{-2}$ on average on the micrographs. The accuracy of the numbers is limited by the contrast, depending on the micrograph. The standard deviations $\Delta n_{CNDs}$ for both average numbers are very high (Supplementary Table 2), with the mean value of both standard deviation being 0.68. This mean value for $\Delta n_{CNDs}\langle n_{CNDs}\rangle^{-1}$ would correspond to the uncertainty in concentration determination $\Delta C_{CND} C_{CND}^{-1} = 0.68$, underlining that the concentration determination with TEM is not feasible. Assuming a homogeneous coating of the whole TEM grid (area $= 7.3 \times 10^6\,\mu m^2$) with these densities, one can estimate $2.4 \times 10^{10}$–$2.8 \times 10^{10}$ CNDs in the dried 2 μL that were drop-casted onto the grids. This would correspond to $c_{CNDs} \approx 10$–20 nM solutions. As expected, this value is several orders of magnitude off the theoretical value, underlining that it is not feasible to obtain a meaningful CND concentration based on TEM analysis.

*Concentration determination by using the nanoparticle tracking analysis (NTA).* Nanoparticle tracking analysis (NTA) was performed with a NanoSight LM10 (Malvern Panalytical) operated with a 405 nm laser. R-CND solution were diluted to $c_{CNDs} = 18\,\mu M$ (see the respective section in "Methods") and S-CNDs to $c_{CNDs} = 5.5\,\mu M$. As can be observed in Supplementary Fig. 5, only large aggregates are tracked by the system for both samples. The main population of CNDs is too small and scatters too weakly to be discernible with this technique. The CND concentrations (which are in fact aggregate concentrations) determined with NTA were $n_{CNDs} \approx 3.4 \times 10^7$ CNDs $mL^{-1}$ for R-CNDs and $n_{CNDs} \approx 1.0 \times 10^7$ CNDs $mL^{-1}$ for S-CNDs. This corresponds to concentrations in the femtomolar range, underlining that CNDs cannot be measured with NTA but also that the number of aggregates in the CND solutions seems negligible. Note that presence of agglomerates to a large extent can be ruled out by the FCS measurements shown in the respective section in "Methods".

*Other physicochemical characterization data.* Additional standard characterization of the CNDs is provided in the form of Fourier-transform Infrared (FT-IR) spectra (KBr), shown in Supplementary Fig. 2, electronic circular dichroism (ECD) spectra shown in Supplementary Fig. 1 and X-ray photoemission spectroscopy (XPS) shown in Supplementary Fig. 3. FT-IR spectra were recorded on a Perkin Elmer 2000 spectrometer. ECD spectra were measured on a Jasco J-815. XPS spectra were measured on a SPECS Sage HR 100 spectrometer.

**Influence of cell culture medium on the properties of the CNDs.** While the characterization in the respective section about CND properties in "Methods" was carried out in water, uptake experiments of the CNDs took place in cell culture

medium. Thus, it needed to be tested how the presence of cell culture medium affects the properties of the CNDs. For this, 400 µL of CND solutions ($C'_{\text{CNDs}}$ = 200 µg mL$^{-1}$) were mixed with the same volume of either (i) Milli-Q water (as the negative control), (ii) RPMI (Roswell Park Memorial Institute) 1640 medium without phenol red (Thermofisher, USA) supplemented with 10% fetal bovine serum (FBS, Biochrom, UK), and (iii) RPMI 1640 medium without phenol red without serum. Thus, the final CND concentration was $C'_{\text{CNDs}}$ = 100 µg mL$^{-1}$. After different incubation times of $t$ = 0, 24, and 48 h, the CND solutions were characterized in a UV-Kuevette, ZH 8.5 mm Deckel (Sarstedt, Germany) with UV-Vis absorption spectroscopy and with fluorescence spectroscopy ($\lambda_{\text{ex}}$ = 405 nm). In addition, the hydrodynamic diameters $d_\text{h}$ of the CNDs in the different media were measured by dynamic light scattering (DLS, Malvern NANO ZS, England)[60]. In Supplementary Figs. 8, 9 and 32 the absorption and fluorescence spectra of the three different types of CNDs are shown. In all three cases there is a slight fluorescence increase of the different types of CNDs after incubation in particular in serum containing RPMI 1640 medium. At this increase is similar, fluorescence intensities of the three different types of CNDs ($R$-CNDs, $S$-CNDs, and N-CNDs) can be also directly compared when the CNDs are exposed to cell culture medium.

The hydrodynamic diameters $d_\text{h}$ of the CNDs after incubation with the different medium were measured after different time points by DLS. Data are shown in Supplementary Fig. 33. Due to the very small size of the CNDs and in the case of RPMI 1640 medium supplemented with 10% FBS due to the presence of proteins of similar size as the CNDs, the DLS values are unreliable and are not further interpreted in this study. In fact, the hydrodynamic diameter as detected in the plain media without proteins (yellow bars; water and RPMI 1640 medium without serum) most likely correspond to dust. The increase hydrodynamic diameters as detected in the serum-supplemented medium (yellow bars, RPMI 1640 medium with FBS) originate from serum proteins. Presence or absence of the CNDs (red and blue versus yellow bars) does not change the results, which demonstrates that it is not the CNDs which are detected here with DLS, which is due to their tiny size. Hydrodynamic radii $r_\text{h} = d_\text{h} \, 2^{-1}$ of the CNDs were instead measured with fluorescence correlation spectroscopy (FCS, see the respective section in "Methods"), where only the CNDs and CND−protein complexes, but not the free proteins provide signal.

**Protein adsorption on CNDs.** To explore whether the chiral surface of CNDs has an impact on protein adsorption, the interaction of the CNDs with different proteins, human serum albumin (HSA, CAS No. 70024-90-7, Sigma Aldrich, Germany), transferrin human (Tf, CAS No. 11096-37-0, Sigma-Aldrich, Germany), and alpha-2-macroglobulin (α2M, SRP6314, Sigma-Aldrich, Germany) was investigated with fluorescence correlation spectroscopy (FCS)[39–41,61–65]. Measurements were carried out in a Confocal Light Scanning Microscope (CLSM) (LSM 880, Zeiss, Germany) with a Zeiss PlaN-Apochromat ×40/1.0 Water DIC (WD: 2.5 mm) objective with integrated FCS set-up (Zeiss). FCS studies were conducted with two solvents either in filtered Milli-Q water or in phosphate-buffered saline (PBS, Gibco, Invitrogen, Belgium). For measurements, proteins at different concentration were mixed with CNDs in either PBS or water, leading to a final variable protein concentration $c_P$ (P = HSA, Tf, α2M) and a fixed CND concentration $C_{\text{CND}}$ = 10 µg mL$^{-1}$ for batch #1 and 50 µg mL$^{-1}$ for batch #2 and batch #3. Before measurements all samples were incubated for 15 min and were then loaded to 35 mm petri dishes with glass bottom (Cat.No: 81218-200, ibidi GmbH, Germany) and were immediately covered by a cover glass (Product Code.10474379, Carl Zeiss™, Germany) with a thickness of 0.17 mm ± 0.005 mm. Subsequently, the lid of the glass bottle dish was assembled before FCS measurement. It is worth to mention, that the glass petri dish and the cover slide were continuously used through all measurements to exclude any possible deviation from their thickness, which could have probably resulted in an experimental error (the parameters $\omega_0$ and S of the excitation volume as described below might vary). To make sure that the glass petri dish and cover slide were sufficiently clean and dried before carrying out the next measurement, they were washed by ethanol and Milli-Q water successively, gently wiped with soft tissue paper and dried thoroughly under room temperature (RT) for 5 min. The FCS set-up had to be calibrated. Before carrying out the measurements, the focal volume was calibrated at 488 nm laser excitation with the laser power of 0.5 (at the Zeiss LSM set-up) using a dye with a known diffusion coefficient $D_{\text{Rho}}$ = 414 ± 1 µm$^2$ s$^{-1}$ (Rhodamine 6G)[66]. Experimentally FCS determines diffusion times $\tau_D$ from an autocorrelation function G($\tau$) based on the fluorescence fluctuation of dyes diffusion in and out into the focus of the excitation. Here fluorescence fluctuations were recorded with 100 repetitions of each 10 s. An example of the autocorrelation function as obtained with Rhodamine 6G dissolved in water ($c_{\text{Rho}}$ = 10 nM) is shown in Supplementary Fig. 34. This autocorrelation function was fitted with the following equation, using the FCS module implemented in the Zeiss ZEN software:

$$G(\tau) = \frac{1}{N}\left(1 + \frac{T}{1-T}e^{-\tau/\tau_T}\right)\sum_{i=1}^{M}\frac{f_i}{1 + \tau/\tau_{\text{Di}}}\frac{1}{\sqrt{1 + \tau/\tau_{\text{Di}}\,S^2}} \quad (1)$$

$N$ is the average number of fluorophores within the effective detection volume, i.e. the volume of the excitation focus. $M$ is the number of different fluorescent components in solution (e.g. if a mix of different fluorophores would be analyzed). In the present case $M$ = 1, as there is either only Rhodamine 6G, or afterwards just CNDs in solution. $f_i$ determines the contribution of the different fluorescent

components to the autocorrelation function. As here there is only one component $f_1$ = 1. $T$ is the fraction of the fluorescence decay from the triplet state of the fluorescent compound, and $\tau_T$ is the lifetime of the triplet state. For the Rhodamine 6G data shown in Supplementary Fig. 34 the mean of the fit parameters from three measurements were $N$ = 0.15 ± 0.01, $S$ = 4 ± 1, and $\tau_D = \tau_{\text{Rho}}$ = 22.5 ± 0.5. For the CNDs, the contribution of fluorescence from the triplet state was neglected, i.e. $T$ = 0 and $\tau_T$ = ∞. The cross-section of the excitation volume is considered as ellipsoid and $S$ is the ratio of the axis of this ellipsoid (for a sphere it would be $S$ = 1). Thus, the effectively applied fit-function reduced to:

$$G(\tau) = \frac{1}{N}\frac{1}{1 + \tau/\tau_D}\frac{1}{\sqrt{1 + \tau/\tau_D\,S^2}} \quad (2)$$

By assuming the excitation volume as ellipsoid (Gaussian ellipsoid approximation), the diffusion time $\tau_D$ related to the corresponding diffusion coefficient $D$ via the width of the excitation volume $\omega_0$ by:

$$\tau_D = \frac{\omega_0^2}{4D} \quad (3)$$

Using the experimentally determined value of the diffusion time $\tau_{\text{Rho}}$ of Rhodamine 6G and the literature value of its diffusion constant $D_{\text{Rho}}$, the width of the excitation volume was calculated to be:

$$\omega_0 = \sqrt{4D_{\text{Rho}}\tau_{\text{Rho}}} = 0.193\,\mu\text{m} \quad (4)$$

By knowing $\omega_0$ as determined for the applied experimental conditions, measured diffusion times $\tau_D$ could be related to diffusion constants $D$, also for the CNDs. In the following these measurements were applied to the different CND samples (different CNDs with different protein concentrations) and for each data point three independent measurements were carried out. In Supplementary Fig. 35 the data for $S$-CNDs (batch #1) as exposed to different concentrations of HSA in PBS are presented. The corresponding diffusion times $\tau_D$ and diffusion coefficients $D$ as determined from the fit as provided in Supplementary Table 11. From the diffusion coefficients $D$ the corresponding hydrodynamic radii were calculated according to the Stokes−Einstein equation:

$$r_h = \frac{k_B T}{6\pi\eta D} \quad (5)$$

$k_B = 1.38 \times 10^{-23}$ J K$^{-1}$ is the Boltzmann constant, $T$ = 298.15 K room temperature, and $\eta$ is the solution viscosity. The solution viscosity was assumed to depend linearly on the protein concentration according to:

$$\eta = \eta_0\left(\eta_i C_P + 1\right) \quad (6)$$

The protein mass concentration $C_P$ relates to the molar protein concentration $c_P$ by the molar mass of the protein MW(P). $\eta_0$ = 0.89 mPa × s is the viscosity of PBS, which is assumed to be the viscosity of water (at room temperature). $\eta_i$ is the intrinsic viscosity of the proteins. Hereby the following values were used: HSA: $M_W$(HSA) = 66.5 kDa, $\eta_i$ = 4.2 cm$^3$ g$^{-1}$; Tf: $M_W$(Tf) = 80 kDa, $\eta_i$ = 4.4 cm$^3$ g$^{-1}$; α2M: $M_W$(α2M) = 725 kDa[40]. In Supplementary Table 11 the conversion of diffusion coefficients $D$ into hydrodynamic radii $r_h$ is demonstrated. The resulting hydrodynamic radii $r_h$ versus protein concentrations $c_P$ are listed of the different types and batches of CNDs as recorded in water in Supplementary Fig. 36 and as recorded in PBS in Supplementary Fig. 10. In the following, all further discussion will be based on the results obtained in PBS. The data recorded in PBS with HSA show a saturation of the hydrodynamic radius, e.g. at high protein concentration the CND surface is completely saturated with proteins and thus the hydrodynamic radius does not increase further with raising concentrations. The behavior can be fitted with the Hill model[39]. For this, the $r_h(c_{\text{HSA}})$ curves shown in Supplementary Fig. 10 were fitted with the following equation:

$$r_h(c_{\text{HSA}}) = r_h(0)\sqrt[3]{1 + \frac{V_{\text{HSA}}}{V_{\text{CND}}}N_{\text{max}}\frac{1}{1 + \left(\frac{K_D}{c_{\text{HSA}}}\right)^n}} \quad (7)$$

$V_{\text{HSA}}$ is the volume of one HSA molecule. Assuming a triangular prism shape with 8.4 nm side length and 3.2 nm height we used $V_{\text{HSA}}$ = 96 nm$^3$ [39]. $V_{\text{CND}}$ is the volume of one CND, considering the CNDs as spheres, and the radius $r_h(0)$ obtained from the FCS of the control sample, no protein:

$$V_{\text{HSA}} = \frac{4}{3}\pi r_h(0)^3 \quad (8)$$

$r_h(0)$ is an experimentally determined value and is enlisted in Supplementary Table 5. The fit function had the following free fit parameters: $r_{h,0}$, $N_{\text{max}}$, $K_D$, and $n$. $r_{h,0}$ is the value from the fit for the hydrodynamic radius of the CNDs with no adsorbed proteins, i.e. $r_h(c_{\text{HSA}} \ll K_D)$. $r_{h,0}$ is a fit parameter, $r_h(0)$ is an experimentally determined value. $N_{\text{max}}$ is the number of HSA molecules bound per CND in saturation (i.e. $c_{\text{HSA}} \gg K_D$), $K_D$ is the dissociation coefficient, and $n$ is the Hill coefficient[39]. The resulting fit values from the curves shown in Supplementary Fig. 10 are presented in Supplementary Table 5.

**Cell culture techniques.** Two cell lines were used in this study: THP-1 monocytes and HeLa cells. The human monocytic leukemia cell line THP-1 (ATCC® TIB-202™) was obtained from American Type Culture Collection (ATCC, Manassas, VA, USA). THP-1 cells were cultured in suspension in RPMI 1640 medium

(Sigma-Aldrich, #61870010) containing 10% heat inactivated fetal bovine serum (FBS, Biochrom, UK), 1 mM sodium pyruvate (Sigma-Aldrich, #S8636), 0.05 mM β-mercaptoethanol (Sigma-Aldrich, #M3148), 100 U mL$^{-1}$ penicillin and 100 μg mL$^{-1}$ streptomycin (P/S, Sigma-Aldrich, Germany) in a humidified incubator at 37 °C and 5% $CO_2$. For experimental usage, Phorbol 12-myristate 13-acetate (PMA, Sigma-Aldrich, #P1585) was applied to the THP-1 monocyte (cell passage less than 30) with a dosage of 150 nM[14] for 3 days, inducing the differentiation from THP-1 monocytes to THP-1 macrophages. After stimulation, the THP-1 macrophages were exposed to the CNDs for in vitro uptake, toxicity, and colocalization studies (Supplementary Fig. 37). The human cervix cell line HeLa cells were purchased from ATCC and maintained in Dulbecco's modified Eagle's medium (DMEM, Thermofisher, USA) supplemented with 10% fetal bovine serum (FBS, Biochrom, Germany) and 100 U mL$^{-1}$ penicillin/streptomycin (P/S, Fisher Scientific, Germany) at 37 °C and 5% $CO_2$, until desired confluence was reached, before adding the CNDs.

**Cell viability assays.** The cell viability of THP-1 derived macrophages and HeLa cells after exposure to CNDs was evaluated by the resazurin assay[50,67]. In viable cells there is a metabolic reduction of the non-fluorescent resazurin to the highly fluorescence of resorufin, and thus this fluorescence is assumed to be proportional to the number of living cells. In case of the addition of toxic materials the number of living cells will be decreased. As cell viability $V$ the percentage of living cells in reference to a sample with untreated control cells is defined[50]. Before evaluating the biocompatibility of the CNDs, several tests were performed to exclude the possibility that the fluorescent CNDs would interfere with the resazurin assay. In a first step, we investigated possible interference effects of CNDs with resazurin (i.e. without exposing the CNDs to cells), analyzing the concern that the CNDs alone could trigger the conversion of resazurin to resorufin. Briefly, several solutions were prepared as follows, including $H_2O$, RPMI 1640 medium supplemented with 10% FBS, 0.025 mg mL$^{-1}$ resazurin solution in RPMI 1640 medium with 10% FBS, $C'_{CND} = 200$ μg mL$^{-1}$ of $R$-CNDs or $S$-CNDs in $H_2O$, $C'_{CND} = 200$ μg mL$^{-1}$ of $R$-CNDs in 10% FBS supplemented RPMI 1640 medium containing 0.025 mg mL$^{-1}$ resazurin (Sigma Aldrich, USA). Then 100 μL of each solution was loaded to a 96-well plate (Sarstedt, Germany) with 0.34 cm$^2$ growth area per well and the fluorescence spectra of each well was collected from 570 to 620 nm by a fluorimeter (Fluorolog-3, Horiba Jobin Yvon, USA) with excitation at 560 nm (Supplementary Fig. 38). Data show that the addition of CNDs to resazurin did not trigger fluorescence (i.e. conversion of resazurin to resorufin). In a second step, we investigated the interference effect of the fluorescence of the CNDs, which might interfere with the fluorescence of resorufin. THP-1 monocytes were seeded at a density of 34,000 cells/well with the medium volume $V_{medium} = 0.136$ mL per well in a 96-well plate with 0.34 cm$^2$ growth area per well and were differentiated to macrophages within in 3 days (Supplementary Fig. 37). Afterwards, the supernatant was removed and 100 μL of $R$-, $S$- or $N$-CNDs ($C'_{CND} = 200$ μg mL$^{-1}$) diluted in RPMI 1640 medium containing 10% FBS were added to the THP-1 derived macrophages. As control cells were exposed to medium without added CNDs. Cells were further incubated for 24 h in a humidified incubator at 37 °C, 5% $CO_2$. The next day, resazurin salt solution (Sigma Aldrich, USA) at a concentration of 0.25 mg mL$^{-1}$ was mixed with 10% FBS supplemented RPMI 1640 medium at a volume ratio of 1:10 as resazurin working solution. The control cells which had not been exposed to CNDs, the cells in each well were washed with 100 μL of phosphate-buffered saline (PBS, Gibco, Invitrogen, Belgium) and 100 μL of resazurin working solution was added. The CND-treated cells remained unchanged. Cells were further incubated at 37 °C and 5% $CO_2$ for 4 h. After this the fluorescence spectra of the different wells were collected from 570 to 620 nm at 560 nm excitation by a fluorimeter (Fluorolog-3, Horiba Jobin Yvon, USA; Supplementary Fig. 38b). As shown in Supplementary Fig. 38b, the significant fluorescence of resorufin from cells which have been treated with resazurin. The fluorescence from cells which had been treated with only CNDs was negligible in comparison to the resorufin fluorescence. This rules out interference of the viability test with the intrinsic CND fluorescence. For the viability tests slightly different protocol were used for the THP-1 monocytes and the HeLa cells. The THP-1 monocytes were seeded at a density of 34,000 cells/well with the medium volume $V_{medium} = 0.136$ mL per well in a 96-well plate with 0.34 cm$^2$ growth area and were differentiated to macrophages. On the fourth day, the supernatant in each well was removed and then CND solution (i.e. CNDs dispersed in medium with or without FBS supplement; $V_{medium} = 0.136$ mL) with a series of different concentrations was added for 24 or 48 h. In contrast, HeLa cells were seeded in 96-well plates with 0.34 cm$^2$ growth area at a density of 7500 cells/well in 0.1 mL DMEM medium supplemented with 10% FBS. On the following day, the old medium was removed then CND solution ($V_{medium} = 0.1$ mL; with or without FBS supplement) with a series of different concentrations was added for 24 or 48 h. For both cell lines, after the incubation time, phosphate-buffered saline (PBS, Gibco, Invitrogen, Belgium) ($V_{PBS} = 0.1$ mL) was used to wash the cells once, then 100 μL of resazurin working solution was added and further incubated for 4 h at 37 °C. The resazurin working solution was prepared by diluting the resazurin salt solution (Sigma Aldrich, USA) at a concentration of 0.25 mg mL$^{-1}$ ten times with 10% FBS supplemented medium (RPMI 1640 in case of THP-1 derived

macrophages and DMEM in the case of HeLa cells). Afterwards, the fluorescence spectra of each well were collected from 570 to 620 nm with an excitation of 560 nm as described above. Subsequently, Matlab software was used for data analysis based on the fluorescence intensity at 590 nm, which was considered proportional to the number of living cells. The viability $V$ represent the fluorescence intensity of cells treated with CNDs normalized to the fluorescence intensity of untreated control cells. All experimental conditions were recorded from triplicate independent experiments. As shown in Supplementary Figs. 39 and 40, both cell lines maintained high viability at different exposure doses of CNDs ranging from $C'_{CND}$ 0.488 μg mL$^{-1}$ to 1000 μg mL$^{-1}$ for the different incubation times (24 and 48 h) in cell culture medium supplemented with or without 10% FBS. These results are consistent with the other biocompatibility tests regarding CNDs[68].

**Time- and dose-dependent uptake studies based on flow cytometry**
*Time-dependent uptake of CNDs by HeLa cells.* Uptake of the different CNDs by HeLa cells was investigated. Firstly, HeLa cells were seeded at a density of 40,000 cells well$^{-1}$ with 10% FBS contained DMEM medium of volume $V_{medium} = 1$ mL per well in 24-well plates (Sarstedt, Germany) with 1.9 cm$^2$ seeding area per well. On the next day, the medium in each well was removed and then CNDs diluted in DMEM medium supplemented with 10% or 0% FBS ($V_{medium} = 0.5$ mL) were added to the HeLa cells for specific time points (1, 3, 6, 24 and 48 h) at a concentration of $C'_{CNDs} = 400$ μg mL$^{-1}$. After the exposure time, cells were washed three times with 0.5 mL cold PBS, detached by addition of 0.05% trypsin-EDTA, isolated by centrifugation at 300 × g for 5 min, and finally re-suspended in 0.3 mL cold PBS for flow cytometer analysis (BD LSRFortessa™, BD Biosciences, US). The CND fluorescence signal $I$ within each cell was collected with the flow cytometer with a 450/50 nm bandpass filter upon 405 nm excitation. 10,000 gated cells were counted and analyzed for each sample. Then, the Flowjo software was used to analyze the flow cytometry data. The recorded mean CND fluorescence per cell $I$ was then background-corrected by subtracting the fluorescence of control cells which had not been exposed to CNDs, leading to the background-corrected mean CND fluorescence per cell:

$$\frac{I^*}{I^*(C_{CNDs})} = I(C'_{CNDs}) - I(C'_{CNDs} = 0) \tag{9}$$

Finally, as the different CNDs at the same concentration $C'_{CND}$ have a different fluorescence the correction factor X as determined in Supplementary Table 4 was taken into account:

$$I'(C'_{CNDs}) = I^*(C'_{CNDs}) \text{ for } S - \text{CNDs} \tag{10}$$

$$I'(C'_{CNDs}) = X_{S/R} I^*(C'_{CNDs}) \text{ for } R - \text{CNDs} \tag{11}$$

$$I'(C'_{CNDs}) = X_{S/N} I^*(C'_{CNDs}) \text{ for } N - \text{CNDs} \tag{12}$$

These effective fluorescence intensities per cell $I'$ were the values which were further compared. In Supplementary Fig. 23 the raw data of the flow cytometry measurements are shown. In Supplementary Fig. 22 the mean fluorescence intensity per cell data as extracted from those raw data are plotted. As additional parameter also the percentage of HeLa cells which had endocytosed CNDs $P_{cell}$ was calculated from the flow cytometer data (Supplementary Fig. 23) by the Flowjo software. Here a fluorescence threshold was set to distinguish cells with fluorescence from internalized CNDs from the autofluorescence of cells. The gating strategy is shown in Supplementary Fig. 41. The results from $n = 3$ experiments are displayed in Supplementary Fig. 42.

*Time- and dose-dependent uptake of CNDs by THP-1 derived macrophages.* Apart from HeLa cells we selected THP-1 derived macrophages as cell model. Macrophages are important cells of the immune system, capable of distinguishing pathogens such as bacteria, cellular debris and foreign entities[69,70]. Thus, they might be in particular sensitive concerning the surface properties of CNDs (such as different chirality of the CNDs) in regard of CND endocytosis. Briefly, THP-1 monocytes were seeded at a density of 100,000 cells/well with 10% FBS containing RPMI 1640 medium with volume $V_{medium} = 0.4$ mL per well in 48-well plates (Sarstedt, Germany) with 1 cm$^2$ growth area per well and they were differentiated into THP-1 derived macrophages within 3 days (see the respective section in "Methods"). Then, the supernatant was removed, and CND solution ($V_{medium} = 0.4$ mL) was added to the THP-1 derived macrophages at final concentrations ranging from $C'_{CNDs} = 50$ μg mL$^{-1}$ to 400 μg mL$^{-1}$ for different time points $t$, including 1, 3, 6, 24 and 48 h in RPMI 1640 medium supplemented with 10% or 0% FBS. After the different incubation time points, cells were washed three times with 0.5 mL cold PBS, detached by 0.05% trypsin-EDTA solution (Thermofisher, USA), isolated by centrifugation at 300 × g for 5 min, and were then re-suspended in 220 μL cold PBS. Subsequently, the re-suspended cells were analyzed by flow cytometry in the same way as described in the respective section in "Methods" for the HeLa cells. In the same way background subtraction and adjustments for the different fluorescence intensities of the CNDs was performed. The results for different incubation concentrations $C'_{CNDs}$ and incubation times are provided in Supplementary Figs. 25 and 43.

**Time-dependent uptake studies based on confocal microscopy**. Uptake of CNDs by THP-1 derived macrophages was also quantified by confocal laser scanning microscopy (CLSM; LSM 510, Zeiss, Germany) with a Plan-Apochromat ×63/1.40 Oil DIC M27 objective. THP-1 monocytes were seeded at a density of 75,000 cells per well in complete RPMI 1640 medium with the volume $V_{medium} = 0.3$ mL per well in μ-Slide 8 Wells (ibidi GmbH, Germany) with 1 cm² growth area per well, and were differentiated into THP-1 derived macrophages within 3 days (see the respective section in "Methods"). On the fourth day, the medium was removed, and CND solution ($V_{medium} = 0.3$ mL) was added to cells at a concentration of $C'_{CNDs} = 400$ μg mL$^{-1}$ in RPMI 1640 medium supplemented with 10% or 0% FBS subsequently. At specific time points $t$ similar to those chosen for flow cytometry experiments (see the respective section in "Methods"), cells were imaged by CLSM using a 405 nm laser as the excitation source and a LP 420 nm long pass filter for recording the fluorescence emission. Representative images are shown in Supplementary Figs. 52 and 53. After obtaining the CLSM images at each time point, quantitative analysis of cellular CND uptake was performed by utilizing a combination of free open-source software. In a first step, the images as obtained from the Zeiss microscope software were converted to TIFF format utilizing Matlab software. In a second step, Adobe photoshop CS6 was used to manually denote the perimeter of the cells. In a third step, the sum of the fluorescence intensities of all pixels belonging to a cell was calculated by the image analysis software Cellprofiler v2.2.0, and converted to the mean fluorescence per cell by dividing the summed up pixel intensities by the number of fluorescent cells[54,71]. The intensity values were then corrected by the fluorescence difference between the different CND sample according to Supplementary Table 4:

$$I'(C'_{CNDs}) = I^*(C'_{CNDs}) \text{ for } S-CNDs \qquad (13)$$

$$I'(C'_{CNDs}) = X_{S/R} I(C'_{CNDs}) \text{ for } R-CNDs \qquad (14)$$

$$I'(C'_{CNDs}) = X_{S/N} I(C'_{CNDs}) \text{ for } N-CNDs \qquad (15)$$

More than 200 cells in at least 20 images from three independent experiments were analyzed for each time point ($n = 3$). No background correction was performed, as the background fluorescence was low. The results are shown in Supplementary Fig. 12 and Supplementary Table 8. In addition, from the microscopy data exemplary shown in Supplementary Figs. 52 and 53 the percentage $P_{cell}$ of cells which had internalized CNDs was determined. Data are shown in Supplementary Fig. 54. This counting was performed manually, which was possible due to the low background. To achieve a low background first a control cell which had not been exposed to CNDs was imaged and the parameters of the confocal microscope (e.g. laser power and pinhole) were adjusted in a way that no fluorescence could be observed by the naked eye in the fluorescence images. With these settings then the cells with internalized CNDs were recorded.

**Colocalization of CNDs with intracellular organelles**

*Colocalization studies of mitochondria or lysosome and CNDs.* Colocalization studies of internalized CNDs with cell organelles, i.e. mitochondria and lysosomes, were carried out for THP-1 derived macrophages and for HeLa cells using Confocal Laser Scanning Microscopy (CLSM) (LSM 510, Zeiss, Germany) with a Plan-Apochromat ×63/1.40 Oil DIC M27 objective. Firstly, THP-1 monocytes were seeded at a density of 75,000 cells per well with complete RPMI 1640 medium volume $V_{medium} = 0.3$ mL supplemented with PMA at a concentration of 150 nM in μ-Slide 8 Wells (ibidi GmbH, Germany) with 1 cm² growth area per well. After 72 h incubation time in a cell culture incubator, the THP-1 monocytes had been differentiated into THP-1 derived macrophages. In the case of HeLa cells, 12,000 cells were seeded per μ-Slide 8 Well with the complete DMEM medium of volume $V_{medium} = 0.3$ mL per well and were cultured in a cell culture incubator at 37 °C in 5% CO₂ overnight. After this, for both cells type the previous medium was replaced with the $V_{medium} = 0.3$ mL CNDs dispersed in RPMI 1640 and DMEM medium supplemented with 10% or 0% FBS at a concentration of $C'_{CNDs} = 400$ μg mL$^{-1}$ for THP-1 derived macrophages and HeLa cells, respectively. After 24 or 48 h incubation time, mitochondria and lysosome were labeled with corresponding staining reagents as described in the following.

*Immunostaining procedures.* For mitochondrial staining, MitoTracker$^R$ Deep Red$^{FM}$ (Catalog No.: M22426, ThermoFisher Scientific)[72] was used to specifically label the mitochondria. Briefly, cells were washed three times with 200 μL PBS and then 200 μL of pre-warmed (37 °C) MitoTracker$^R$ Deep Red$^{FM}$ in complete RPMI 1640 medium at a concentration of 400 nM was added and cells were further incubated in the incubator for 30 min at 37 °C. Afterwards, the staining solution was replaced with fresh pre-warmed RPMI 1640 medium without phenol red (Catalog No.: 11835030, ThermoFisher Scientific) and cells were observed using CLSM. A laser diode emitting at 405 nm and a bandpass emission filter BP 420–480 nm were used to visualize the CNDs. A helium–neon laser of 633 nm and long pass filter LP 650 nm were used for recording the fluorescence of Mito-Tracker$^R$ Deep Red$^{FM}$. For lysosome staining, LysoTracker™ Green DND-26 (Catalog No: L7526, ThermoFisher Scientific) was selected as the lysosomal marker[73–75]. Firstly, cells were washed three times with 200 μL PBS. Subsequently, 200 μL of pre-warmed (37 °C) LysoTracker™ Green DND-26 at a concentration of 1 μM in complete RPMI 1640 medium was added and cells were further incubated

at 37 °C in 5% CO₂ for 30 min prior to imaging with CLSM. The excitation laser and emission collection setups were the same for the CNDs as that used in the colocalization studies of mitochondria and CNDs. An argon laser of 488 nm together with the BP 505–530 nm bandpass filter were used for observing the fluorescence of LysoTracker™ Green DND-26.

*Calculation of Manders' coefficients from the colocalization data.* Based on the CLSM images, colocalization was quantified by quantitatively calculating Manders' coefficients $m_1$ and $m_2$, which are indicators of the overlap degree between pixels from two different fluorescence channels ranging from 0 to 1[73,76,77]. To achieve this purpose, Matlab and Cellprofiler v2.2.0 were used to calculate Manders' coefficients[54,77,78]. Briefly, the Matlab software was firstly used to subtract the background from the 8-bit grayscale TIFF images. Secondly, Cellprofiler v2.2.0 was used to identify pixels belonging to cells. Then the colocalization of CNDs and mitochondria or lysosomes for all pixels corresponding to cells was calculated. Thirdly, the below given equations were used to calculate Manders' coefficients using Matlab. Hereby $m_1$ is the percentage of blue fluorescent pixels (i.e. CNDs) that overlapped with red or green fluorescent pixels (i.e. mitochondria or lysosomes). $m_2$ is the percentage of red or green fluorescent pixels (i.e. mitochondria or lysosomes) which overlap with blue fluorescent pixels (i.e. CNDs).

$$m_1 = \frac{\sum_i I(B)_{i,coloc}}{\sum_i I(B)_i} \qquad (16)$$

$$m_2 = \frac{\sum_i I(R)_{i,coloc}}{\sum_i I(R)_i} \text{ or } m_2 = \frac{\sum_i I(G)_{i,coloc}}{\sum_i I(G)_i} \qquad (17)$$

Hereby $i = 1 \ldots N$ denotes all $N$ pixels which belong to cells. $I(B)_i$, $I(R)_i$, and $I(G)_i$ are the fluorescence intensities $I$ at pixel $i$ as obtained from the blue, red, and green channels of the fluorescence images. $I(B)_{i,coloc}$, $I(R)_{i,coloc}$, and $I(G)_{i,coloc}$ are the fluorescence intensities $I$ from the blue, red, and green channel only for pixels $i$ where there is also green/red, blue, and blue fluorescence. $m_1$ or $m_2 = 0$ represents no overlap, while 1 denotes complete overlap of the fluorescence channels. As expected, the CNDs were largely localized inside lysosomes, and did not co-locate to a large amount with mitochondria. Selected fluorescence images and the resulting Manders' coefficient calculations are shown in Supplementary Figs. 15 and 56–66.

**Studies about the uptake pathway**

*Distinguishing CNDs adherent to the cell membrane from endocytosed CNDs.* Flow cytometry based on standard fluorophores cannot distinguish trivially between CNDs adherent only on the outer cell membrane from actually endocytosed CNDs[79]. Both scenarios however can be distinguished using confocal microscopy. To confirm that the majority of fluorescence signal from CNDs associated with cells originates from internalized CNDs z-stacks were recorded[80] with CLSM (LSM 510, Zeiss, Germany) with a PlaN-Apochromat ×63/1.40 Oil DIC M27 objective. Herein, Calcein-AM (Molecular Probes) was applied to label living cells (i.e. cells with esterase activity)[50], through enzymatic transformation from the non-fluorescent calcein AM to the highly fluorescent calcein[81]. Esterase activity only happens inside cells, and thus is a good label for the volume of the cell. In case the blue fluorescence CNDs overlapped with the green calcein AM fluorescence they can be considered internalized. The experiment was performed in the following way. On the first day, THP-1 monocytes were seeded at a density of 75,000 cells per well with complete RPMI 1640 medium ($V_{medium} = 0.3$ mL) in μ-Slide 8 Wells with 1 cm² growth area per well and were differentiated into THP-1 derived macrophages for 3 days. Then, the supernatant was removed and afterwards the CND ($V_{medium} = 0.3$ mL) dispersed in RPMI 1640 medium with or without serum were added to cells at a concentration of $C'_{CNDs} = 400$ μg mL$^{-1}$ for 24 h. Afterwards, cells were washed three times with PBS (200 μL each) followed by adding 200 μL pre-warmed (37 °C) Calcein AM (one component of the LIVE/DEAD™ Viability/Cytotoxicity Kit, Catalog No: L3224, Thermofisher Scientific) diluted in complete RPMI 1640 medium at a concentration of 1.3 μM. The cells were further incubated with the calcein staining solution at 37 °C for 30 min. Afterwards, the cells were washed with 200 μL PBS once and 300 μL of fresh pre-warmed RPMI medium without phenol red was added prior to imaging by confocal microscopy using z-stack analysis (Supplementary Figs. 67 and 68)[80]. A laser diode emitting at 405 nm and a bandpass emission filter BP 420–480 nm were used for the excitation and emission collection of CNDs. An argon laser of 488 nm together with a bandpass emission filter BP 505–550 nm were used to visualize the fluorescence of calcein. In Supplementary Figs. 67 and 68, cross-sections through cells as indicated by the red and green solid lines are shown. These data clearly show that the blue fluorescence emitted from the CNDs originates from inside the cells, and thus corresponds to endocytosed CNDs.

*Uptake of CNDs by THP-1 derived macrophages under the presence of inhibitors.* The endocytosis pathway of CNDs can be investigated by evaluating the inhibition of certain internalization pathways by pharmacological/chemical inhibitors associated with[82]. In our case, we investigated the cellular uptake pathway of CNDs using the following chemical inhibitors: (i) Nocodazole[83] (an inhibitor of endocytosis of lager nanoparticles[82,84,85], CAS No.31430-18-9, Sigma-Aldrich, Germany), (ii) Bafilomycin A1[84] (an inhibitor of phagocytosis[86,87], CAS No. 88899-55-2, InvivoGen, France), (iii) Amiloride[88] (an inhibitor of micropinocytosis[89,90], CAS No. 17440-83-4, Sigma-Aldrich, Germany), and (iv) Chlorpromazine[88] (an inhibitor

of clathrin-associated endocytosis[89,91], CAS No.69-09-0, Sigma-Aldrich, Germany). While the inhibitors were applied, still cytotoxicity experiments for the above inhibitors were carried out to ensure that at the used concentrations the inhibitors are not toxic. For experiments on the first day, THP-1 monocytes were seeded at a density of 34,000 cells per well with complete RPMI 1640 medium ($V_{medium}=0.136$ mL) in 96-well plates with 0.34 cm$^2$ growth area per well and were differentiated into THP-1 derived macrophages. On the fourth day, the supernatant was removed and then each inhibitor diluted in RPMI 1640 medium supplemented with 10% or 0% FBS at several concentrations was added to the cells for 7 or 25 h. The exposure concentrations of each inhibitor are summarized in Supplementary Table 16. After the incubation time of 7 h or 25 h, the cell viability test was performed following the protocols described in the respective section in "Methods". As shown in Supplementary Fig. 69, the four chemical inhibitors were non-toxic to THP-1 derived macrophages at the used concentration ranges. To ensure that there is also not reduction of cell viability of the THP-1-derived macrophages after exposing to CNDs pre-incubated with the cellular uptake inhibitors, also for this scenario a viability assay was carried out. Briefly, THP-1 monocytes were differentiated into THP-1 derived macrophages with an original seeding density of 34,000 cells per well with complete RPMI 1640 medium volume ($V_{medium}=0.136$ mL) in 96-well plates. After 3 days, the previous medium was substituted with 0.136 mL of fresh RPMI 1640 medium containing 10% or 0% FBS containing optionally Nocodazole, Bafilomycin A1, Amiloride, or Chlorpromazine. The concentrations used for each inhibitor are described in Supplementary Table 17. Afterwards, cells treated with the inhibitors were incubated at 37 °C. Cells cultured at 37 °C without exposure to inhibitors were used as positive controls. Cells cultured at 4 °C (i.e. conditions where there is reduced endocytosis) without exposure to inhibitors were used as negative controls. After 1 h incubation time, the three types of CNDs were directly added at a final concentration of $C'_{CNDs}=400\,\mu g\,mL^{-1}$ and the 96-well plates were further incubated at the original conditions for another 6 h before the viability measurements. As shown in Supplementary Figs. 70 and 71, the exposure of the above inhibitors together with the CNDs did not reduce viability of the THP-1 derived macrophages. Thereafter, the cellular uptake pathway of CNDs in THP-1 derived macrophages was investigated. The experiment was conducted as follows. On the first day, THP-1 monocytes were seeded at a density of 100,000 cells per well with 10% FBS containing RPMI 1640 medium ($V_{medium}=0.4$ mL) in 48-well plates (Sartstedt, Germany) with 1 cm$^2$ growth area per well. After 3 days, the cells were differentiated into THP-1 derived macrophages. Afterwards, the previous cell culture medium was replaced with fresh RPMI 1640 medium containing 10% or 0% FBS, supplemented optionally with Nocodazole, Bafilomycin A1, Amiloride, or Chlorpromazine at the concentrations described in Supplementary Table 17. The cells treated with the inhibitors were cultured at 37 °C for 1 h. Cells without exposure to inhibitors were cultured at 37 °C as positive controls. Cells without exposure to inhibitors were cultured at 4 °C (i.e. there is reduced endocytosis) as negative control. After the incubation time, the three types of CNDs were directly added at a final concentration of 400 μg mL$^{-1}$ and the plates were further incubated at the original conditions for another 6 h before flow cytometry analysis. The sample collection procedure, flow cytometry setups, and data analysis methods were the same as described in the respective section in "Methods". As shown in Supplementary Figs. 72 and 73, the cellular uptake of the CNDs was blocked at 4 °C, suggesting that the internalization of the CNDs is an energy-dependent process[55]. Presence of Bafilomycin A1 drastically reduced the uptake of CNDs. Thus, phagocytosis will be a major route of uptake of the CNDs[86,87]. In case of the N-CNDs also presence of Chlorpromazine reduced uptake of the CNDs. Thus here also clathrin-associated endocytosis will be a relevant uptake pathway for N-CNDs[89,91]. We point out again that the N-CNDs were partially agglomerated, which might be the reason for this additional pathway, which is most likely not related to the chirality of the CNDs. Addition of nocodazole did not reduce CND uptake, which is understandable as this has been reported to block in particular uptake of larger particles[84,85]. Amiloride caused autofluorescence in the cells, which is why under presence of this inhibitor there is elevated fluorescence also in the presence of CNDs, which however is autofluorescence of the inhibitor.

*Uptake of CNDs by Hela cells under the presence of inhibitors.* The same experiments as done with THP-1 derived macrophages were also carried out for Hela cells. First, the cytotoxicity of CNDs pre-incubated with cellular uptake inhibitors was conducted. For that, 7500 HeLa cells in 0.1 mL complete DMEM medium were seeded in 96-well plates with a growth area of 0.34 cm$^2$ per well at 37 °C overnight. Afterwards, the old medium was removed and HeLa cells were incubated with fresh DMEM medium containing 10% or 0% FBS without or with inhibitors at the concentrations enlisted in Supplementary Table 17. Then, the inhibitor treated cells were incubated at 37 °C. As positive control cells without presence of inhibitors were used. As negative control cells without exposure to inhibitor were cultured at 4 °C instead at 37 °C. After 1 h incubation, the three types of CNDs were added to the cells at a final concentration of $C'_{CNDs}=400\,\mu g\,mL^{-1}$ (i.e. the inhibitors were not removed) and the plates were furtherly cultured at the original conditions (37 °C or 4 °C for the negative control) for another 6 h. Then the resazurin assay was applied. The results shown in Supplementary Figs. 74 and 75 demonstrated that the CNDs and chemical cellular uptake inhibitors did nor reduce the viability of the HeLa cells. In order to analyze the effect of the different inhibitors on the uptake of CNDs by Hela cells the same strategy as for the THP-1 derived macrophages was used. Briefly, 40,000 HeLa cells in 1.0 mL 10% or 0% FBS containing

DMEM medium were seeded into 24-well plates with 1.9 cm$^2$ growth area per well at 37 °C overnight. On the following day, the medium was substituted with fresh DMEM medium supplemented with 10% or 0% serum without or with cellular uptake inhibitors at desired concentrations (Supplementary Table 17). The cells treated with inhibitors (or as positive control without inhibitors) were incubated at 37 °C for 1 h. As negative control cell which have not been treated with inhibitor were cultured at 4 °C for 1 h. After this incubation interval, the three types of CNDs were added to the cells (without removing the medium with the inhibitors) at a final concentration of $C'_{CNDs}=400\,\mu g\,mL^{-1}$. Then the plates were returned back with the original incubation conditions (37 °C or 4 °C for the negative control) for another 6 h before sample collection for flow cytometry analysis. The sample collection, flow cytometry setups and data analysis approach were used as described in the respective section in "Methods". Note that here no correction for the different fluorescence emissions of the different types of CNDs was carried out, i.e. $I$ and not $I'$ is plotted. As can be seen in Supplementary Figs. 76 and 77 incubation at 4 °C blocked CND uptake by cells, indicating that endocytosis of the CNDs is an energy consumed process. Besides, the three types of CNDs were taken up by the HeLa cells largely via a phagocytosis pathway, while N-CNDs could also be endocytosed by the HeLa cells via a clathrin-associated endocytosis pathway, similar to the results obtained for the THP-1 derived macrophages.

**Reporting summary**. Further information on research design is available in the Nature Research Reporting Summary linked to this article.

## Data availability
The data supporting the findings of this study are either provided in the article and its Supplementary Information or available from the corresponding authors upon request.

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

## Acknowledgements
H.Y. and S.M. had fellowships from Chinese Scholarship Council (CSC) and Deutscher Akademischer Austauschdienst (DAAD), respectively. This project was funded by the Cluster of Excellence 'Advanced Imaging of Matter' of the Deutsche For-schungsgemeinschaft (DFG)—EXC 2056—project ID 390715994. Further support was received from the University of Trieste, INSTM, AXA Research Fund, the Spanish Ministry of Science, Innovation and Universities, MICIU (project PID2019-108523RB-I00), the Italian Ministry of Education MIUR (cofin Prot. 2017PBXPN4), the Maria de Maeztu Units of Excellence Program from the Spanish State Research Agency (MDM-2017-0720), and the European Research Council (Advanced Grant agreement 885323). Parts of this work were supported by Fraunhofer Attract (Fraunhofer-Gesellschaft).

## Author contributions
H.Y. and M.C. designed and performed the experiments. M.P., W.J.P. and N.F. conceived and coordinated the project. W.J.P. wrote the manuscript with input from all the authors. L.Đ., F.A. and D.Z. oversaw the research and contributed to the experimental design. S.M. contributed with the FCS measurements. F.S. contributed with the TEM and NTA analyses. M.P., W.J.P. and N.F. oversaw the research and secured the funding.

## Competing interests
The authors declare no competing interests.
