## [Peer Review File · Nature Communications]

REVIEWER COMMENTS

Reviewer #1 (Remarks to the Author):

This manuscript studies some of the physiological differences between S- and R-carbon nanodots; namely their differential protein adsorption that is claimed to depend on the differential chirality. S-CNDs are shown to bind/adsorb less human serum albumin (HSA) than R-CNDs. This translates to a higher extent of CND endocytosis of S-CNDs by THP-1 macrophages compared to R-CNDs. This is an interesting concept; one that I had not considered with respect to the interactions of CNDs with biological molecules and how that translates to eventual biological function. The concept is interesting and could have important implications for the design and implementation of a variety of nanoparticles, especially in vivo. However the issue of the determination of CND concentration used (absorbance normalization) makes it impossible to support the claims of the paper if the CNDs are not truly adjusted to near-equal concentrations. I recommend that this be done before the manuscript can be considered further.

Manuscript needs to be polished for grammar.

For this type of study particularly, where the concentration of the NPs plays a critical role in the observed phenomenon, might it be better to determine the concentration of the CNDs by either counting number of CNDs in a fixed volume by TEM and extrapolating or by laser/Brownian motion counting (such as nanosight)? This seems it would be more accurate method of determining CND concentration and adjusting to equal particle concentrations relative to absorbance measurements and normalization. Therefore, it seems that for the authors to fully support the claims of the paper, it is imperative that the NP concentration be determined using an actual particle counting method and not rely on adjustment to equivalent concentrations by normalizing to equivalent UV-vis absorption.

How does size distribution or polydispersity, as determined by DLS to give the hydrodynamic diameter, within each S or R CND population affect the observed chirality-related phenomenon. Are there implications of this chirality issue for janus-like particles of architecture of alternating stripe-like surface feature?

Overall recommendation: manuscript not suitable for publication. May be suitable after major revisions.

Reviewer #2 (Remarks to the Author):

In the current study, carbon nanodots (CNDs) without/with opposite chirality (R-CNDs and S-CNDs) were synthesized, followed by detailed analysis about the comparison of the concentration of both types of CNDs. The authors also studied protein adsorption quantification in 3 typical proteins and demonstrated that CNDs are very weak binders of those proteins. There was only significantly less HSA adsorption on S-CNDs than on R-CNDs and this difference was associated with different cellular behaviors in THP-1 derived macrophages, which endocytose S-CNDs to a significantly higher extent than R-CNDs. In general, the current study is dealing a critical aspect of physicochemical properties of chirality of nanoparticles and the authors identified some interesting observations due to the differential absorption of chiral molecules and this directly related to different cellular uptake. There are some key issues the authors may need to further address to improve the current manuscript.

1. The overall biological significance of the study needs to be further improved and there is no clear purpose why cellular uptake was investigated and what is the matter.
2. The authors studied only 3 typical proteins and then generated the conclusion. It is recommended to study the serum proteins and if the same less protein binding is observed then

the conclusion is more reliable.

3. The authors studied a few cell lines, one of which is macrophage-like cells. It is more convincing to study other macrophages, such as primary macrophages to confirm the observation.

4. As the authors mentioned that several physicochemical properties entangled together, the authors may discuss more about their results in this context.

5. Furthermore, other pairs of chiral counterparts of NPs are studied? What the general pattern is expected?

Reviewer #3 (Remarks to the Author):

In this paper, carbon nanodots (CNDs) with opposite chirality and achiral nanodots were prepared and their interactions with proteins and cells were studied. Protein adsorption quantification and internalization pathways were investigated painstakingly. This is an interesting research in chiral nanostructures. There are several points need to be clarified.

1. Some related references were missing. For example:

(1) *Angewandte Chemie International Edition*, 2021, 10.1002/anie.202101609.

(2) *Nature Communications* 2017, 8, 2007.

2. Sub-Title 'Co-localization studies of mitochondria of lysosome and CNDs' in SI, there is something wrong with the expression.

3. CNDs used in this study were synthesized according to previously reported methods. I think the basic physicochemical characteristic data should still be provided in supporting information, such as atomic force microscope images, circular dichroism spectra, Fourier-transformed infrared spectroscopy (FT- IR) and X-ray photoelectron spectroscopy (XPS) and so on.

4. In addition, the potential use of chiral CNDs in vivo should be discussed.

Rebuttal to the comments of the 3 reviewers

Please note that during revision an additional author has been added: Florian Schulz, who has carried out the TEM and Nanosight measurements added in the revision phase.

Reviewer #1:

This manuscript studies the some of the physiological differences between S- and R-carbon nanodots; namely their differential protein adsorption that is claimed to depend on the differential chirality. S-CNDs are shown to bind/adsorb less human serum albumin (HAS) than R-CNDs. This translates to a higher extent of CND endocytosis of S-CNDs by THP-1 macrophages compared to R-CNDs. This is an interesting concept; one that I had not considered with respect to the interactions of CNDs with biological molecules and how that translates to eventual biological function. The concept is interesting and could have important implications for the design and implementation of a variety of nanoparticles, especially in vivo. However the issue of the determination of CND concentration used (absorbance normalization) makes it impossible to support the claims of the paper if the CNDs are not truly adjust to near-equal concentrations. I recommend that this be done before the manuscript can be considered further.

Manuscript needs to be polished for grammar.

⇒ We have done this to the best of our abilities.

For this type of study particularly, where the concentration of the NPs plays a critical role in the observed phenomenon, might it be better to determine the concentration of the CNDs by either countin number of CNDs in a fixed volume by TEM and extrapolating or by laser/Brownian motion countin (such as nanosight)? This seems it would be more accurate method of determining CND concentration and adjusting to equal particle concentrations relative to absorbance measurements and normalization. Therefore, it seems that for the authors to fully support the claims of the paper, it is imperative that the NP concentration be determined using an actual particle counting method and not rely on adjustment to equivalent concentrations by normalizing to equivalent uv-vis absorption. How does size distribution or polydispersity, as determined by DLS to give the hydrodynamic diameter, within each S or R CND population affect the observed chirality-related phenomenon.

⇒ We are grateful to the reviewer for his/her comments. In fact, we did not put forward enough our key argument. The CNDs used here do not contain metals and thus can't be quantified with ICP-MS. In addition, they are very small and thus provide little signal. This means that typically used standard concentration determination techniques, such as particle counting with AFM or TEM or NTA, will fail. We have performed measurements with these different techniques and present them now in the Supplementary Information. In fact, data show that error in concentration determination with these techniques is higher than with the optical methods as we have finally used in our study. Thus, we already worked with the best concentration determination in our hands. The key message is that there are certain types of particles for which the concentration is very hard to determine in an accurate way (i.e. small organic NPs with low fluorescence). Thus, as concentration determination is prone to large

errors, only big biological effects can be related to NP concentration-dependent properties. We have added a large section to better explain this in the main manuscript and provide also experimental data for alternative concentration determination methods in the Supplementary Information:

In addition, in order to directly compare the biological impact of NPs with different chirality a metric needs to be defined on how properties of different NPs can be compared at the same concentration. Given the fact that surface coatings modify the molecular weight of NPs {Feliu, 2016 #32955}, it is not the same metric to measure at the same mass concentration or to measure at the same NP number concentration.

Due to the small size and the carbon composition of the CNDs, it is a big challenge to define a reliable metric and thus we will first discuss the different approaches in this regard.

In order to determine number concentrations, i.e. the number of NPs per volume of solution or their molarity (with Avogadro's number being the scaling factor between these two entities), the NPs in a fixed volume of solution need to be counted. For big enough NPs counting can be performed easily with optical microscopy {Parakhonskiy, 2015 #32632}. Due to their small size this however is not possible for the CNDs. In principle, small NPs can be counted by immobilizing them on a surface (optionally with evaporation of the solvent) and by imaging them with high-resolution microscopies, such as atomic force microscopy (AFM) or transmission electron microscopy (TEM). Note that for such single NP imaging the resolution of the microscope given by the refraction limit does not necessarily need to be better than the size of the NPs. By working with strongly diluted solutions statistically each signal comes from an individual NP and agglomerates can be excluded, and thus counting of NPs can be performed without being able to resolve them. However, as in this case the number of NPs per image is low, there is a huge error in the counting statistics. In the case of the CNDs investigated in this study the relative error in counting, which determines the uncertainty in concentration determination is $\Delta C_{\text{CNDs}} C_{\text{CNDs}}^{-1} = 43\%$ (Supplementary Table 1). We performed also counting of the CNDs with TEM, which was complicated by their low contrast due to their carbon composition. As TEM with improved refraction limit allows for resolving of individual CND here higher CND concentrations could be used and thus the number of CND counted per image could be increased. However, here agglomeration of the CNDs on the TEM grid occurred, and the relative error in counting, which describes the uncertainty in concentration determination was determined to be $\Delta C_{\text{CNDs}} C_{\text{CNDs}}^{-1} = 68\%$ (Supplementary Table 2). Another common way for NP counting is nanoparticle tracking analysis (NTA). However, the here used CNDs are below the size limit recommended by the manufacturer of the frequently used Nanosight instrument (the manufacturer Malvern Panalytical recommends NPs >10 nm diameter) and due to their low fluorescence emission intensity individual CND does not provide sufficient signal to be detected. Only rare agglomerates of CND are detected, leading to artificial huge hydrodynamic diameter (Supplementary Figure 5). Thus, for the here used CNDs standard NP counting methodologies cannot be applied due to the huge experimental error.

An often-used alternative method to NP counting for the determination of NP concentrations is mass determination. In case of metal NPs the elemental amount of metal from the NPs and thus their concentration can be conveniently determined for the example with inductively coupled plasma mass spectrometry (ICP-MS) {Talamini, 2017 #34819}. However, ICP-MS is not a convenient method for carbon-based NPs such as the here investigated CNDs.

For this reason, here concentration determination of the CNDs was performed based on their optical properties, i.e. molar extinction coefficient and quantum yield.

Concentration determination by CND counting with atomic force microscopy (AFM)

Diluted CNDs solutions of concentration C_{CNDs} of *R*-CNDs and *S*-CNDs were drop-casted on mica substrates for AFM analysis. Images were acquired by tapping mode AFM (Nanoscope IIIa, VEECO Instruments) on a surface area A_{scan} of $25 \mu\text{m}^2$ (Supplementary Figures 28 and 29). The number N_{CNDs} of CNDs as identified in the image (by looking at the height profiles) was counted for samples prepared by two concentrations per typology of CNDs (*R*- and *S*-) and is displayed in Supplementary Table 1. The respective mean values $\langle N_{\text{CNDs}} \rangle$ and standard deviations ΔN_{CNDs} were then calculated.

Taking together all determined 4 values for $\Delta N_{\text{CNDs}} / \langle N_{\text{CNDs}} \rangle^{-1}$ leads to a mean value of $\Delta N_{\text{CNDs}} / \langle N_{\text{CNDs}} \rangle^{-1} \approx 0.21$. The normalized error in particle counting corresponds to the normalized error in concentration determination $\Delta C_{\text{CND}} / C_{\text{CND}}^{-1}$. Note that the real error even might be higher. This point is better rationalized if we consider the mass per particle to compare the theoretical and calculated number of CNDs per substrate area. Considering the similar average size of CNDs, as reported in our previous work, we may calculate the average particle mass of CNDs considering their sphericity and the density of amorphous carbon ($\rho_C = 3.50 \text{ g cm}^{-3}$). By calculating the expected and measured number of particles on substrate, $\langle n_{\text{CNDs}} \rangle = \langle N_{\text{CNDs}} \rangle A^{-1}$, is thus possible to make a direct comparison of the two quantities.

As reported in Supplementary Table 1, the calculated and expected $\langle n_{\text{CNDs}} \rangle$ values are different by several order of magnitude and this result may be influenced by the drop casting deposition process that could be cause of: (i) no homogeneous distribution of particles on the surface, (ii) formation of aggregates during the solvent evaporation.

The following paragraph, treating the CNDs quantification trough transmission electron microscopy (TEM), further remarks the difficulty on calculating these nanoparticles trough microscopy techniques using an analogous concept.

Supplementary Figure 28. AFM of R-CNDs. Tapping mode AFM ($5.0 \times 5.0 \mu\text{m}$) from a drop-casted aqueous solution of R-CNDs on a mica substrate (scale bar, $1 \mu\text{m}$). The inset is the height profile $h(d)$ along the dashed line.

Supplementary Figure 29. AFM of S-CNDs. Tapping mode AFM ($5.0 \times 5.0 \mu\text{m}$) from a drop-casted aqueous solution of S-CNDs on a mica substrate (scale bar, $1 \mu\text{m}$). The inset is the height profile $h(d)$ along the dashed line.

Supplementary Table 1. Counting of CNDs with AFM.

CND type	Sample number	A_{scan} (μm^2)	C_{CNDs} ($\mu\text{g mL}^{-1}$)	N_{CNDs}	$\langle N_{\text{CNDs}} \rangle$	ΔN_{CNDs}	$\Delta N_{\text{CNDs}} \langle N_{\text{CNDs}} \rangle^{-1}$	Theoretical $\langle n_{\text{CNDs}} \rangle$ (μm^{-2})	Calculated $\langle n_{\text{CNDs}} \rangle$ (μm^{-2})
R-CND	1	25	327	35	41.7	6.1	0.15	1.73×10^6	1.67
R-CND	2	25	327	43					
R-CND	3	25	327	47					
R-CND	4	25	108	18	15.7	2.5	0.16	4.01×10^5	0.63
R-CND	5	25	108	16					
R-CND	6	25	108	13					
S-CND	1	25	363	9	8.7	2.5	0.28	1.20×10^6	0.35
S-CND	2	25	363	11					
S-CND	3	25	363	6					
S-CND	4	25	121	8	7.0	1.7	0.24	4.99×10^5	0.28
S-CND	5	25	121	8					
S-CND	6	25	121	5					

Concentration determination by CND counting with transmission electron microscopy (TEM)

Transmission electron microscopy (TEM) measurements to count CNDs were performed using a Jeol JEM-1011 instrument operating at 100 kV. 2 μL of the according CND solution ($C_{\text{CNDs}}(R\text{-CNDs}) = 2.0 \text{ mg mL}^{-1}$; $C_{\text{CNDs}}(S\text{-CNDs}) = 2.8 \text{ mg mL}^{-1}$) were drop-casted onto a copper grid (400 mesh, diameter 3.05 mm) coated with amorphous carbon. As can be observed from the TEM-micrographs shown in Supplementary Figures 30 and 31, homogeneous coating was not achieved. For the *R*-CNDs different aggregates were observed, whereas for the *S*-CNDs some areas with dispersed CNDs were also found. It is not known, how much of the aggregates form during drying on the TEM grid. On both samples it was possible to differentiate single CNDs, however, due to the limited contrast, an exact determination of CND size was not possible. Based on micrograph analysis with ImageJ we obtained $d_{\text{TEM}} \approx 1.5 \pm 0.4 \text{ nm}$ for *R*-CNDs ($N = 216$ CNDs investigated) and $d_{\text{TEM}} \approx 2.4 \pm 0.9 \text{ nm}$ for *S*-CNDs ($N = 255$ CNDs investigated). These values are compatible with those obtained by AFM (d_{AFM}) and discussed in the main text. We emphasize, however, that due to the limited number of observed CNDs and limited contrast, the diameters as determined with TEM should be considered a rough estimate. We used the determined CND diameters to calculate theoretical concentrations, assuming sphericity. With the density of amorphous carbon $\rho_C = 3.50 \text{ g cm}^{-3}$ (diamond has a similar density of $\rho_C = 3.51 \text{ g cm}^{-3}$) and the weight concentrations ($C_{\text{CNDs}}(R\text{-CNDs}) = 2.0 \text{ mg mL}^{-1}$; $C_{\text{CNDs}}(S\text{-CNDs}) = 2.8 \text{ mg mL}^{-1}$) we obtain a molar concentration of $c_{\text{CNDs}} \approx 540 \mu\text{M}$ for *R*-CNDs and $c_{\text{CNDs}} \approx 180 \mu\text{M}$ for *S*-CNDs. Because of the limited accuracy of CND diameter determination, also these concentrations must be considered as a rough estimate.

In the TEM micrographs of *S*-CNDs we find $\langle n_{\text{CNDs}} \rangle = \langle N_{\text{CNDs}} \rangle A_{\text{scan}}^{-1} \approx 3014 \text{ CNDs } \mu\text{m}^{-2}$ on average (Supplementary Table 2). For *R*-CNDs we find $\langle n_{\text{CNDs}} \rangle \approx 1920 \text{ CNDs } \mu\text{m}^{-2}$ on average on the micrographs. The accuracy of the numbers is limited by the contrast, depending on the micrograph. The standard deviations Δn_{CNDs} for both average numbers are very high (Supplementary Table 2), with the mean value of both standard deviation being

0.68. This mean value for $\Delta n_{\text{CND}s} \langle n_{\text{CND}s} \rangle^{-1}$ would correspond to the uncertainty in concentration determination $\Delta C_{\text{CND}} C_{\text{CND}}^{-1} = 0.68$, underlining that the concentration determination with TEM is not feasible.

Assuming a homogeneous coating of the whole TEM grid (area = $7.3 \times 10^6 \mu\text{m}^2$) with these densities, one can estimate 2.4×10^{10} - 2.8×10^{10} CNDs in the dried $2 \mu\text{L}$ that were drop-casted onto the grids. This would correspond to $c_{\text{CND}s} \approx 10 - 20 \text{ nM}$ solutions. As expected, this value is several orders of magnitude off the theoretical value, underlining that it is not feasible to obtain a meaningful CND concentration based on TEM analysis.

Supplementary Figure 30. TEM of *R*-CNDs. TEM images from dried drop-casted aqueous solution of *R*-CNDs on TEM grids.

Supplementary Figure 31. TEM of S-CNDs. TEM images from dried drop-casted aqueous solution of S-CNDs on TEM grids.

Supplementary Table 2. Counting of CNDs with TEM.

CND type	Sample number	A_{scan} (nm ²)	c_{CNDs} (μM)	N_{CNDs}	n_{CNDs} (μm ⁻²)	$\langle n_{\text{CNDs}} \rangle$ (μm ⁻²)	Δn_{CNDs} (μm ⁻²)	$\Delta n_{\text{CNDs}} \langle n_{\text{CNDs}} \rangle^{-1}$
R-CND	1	1.68×10^4	540	75	4464	1920	1620	0.84
R-CND	2	6.42×10^4	540	141	2196			
R-CND	3	1.79×10^4	540	55	3073			
R-CND	4	4.22×10^4	540	136	3223			
R-CND	5	2.59×10^5	540	108	417			
R-CND	6	2.59×10^5	540	37	143			
R-CND	7	2.59×10^5	540	454	1753			
R-CND	8	1.03×10^6	540	96	93			
S-CND	1	1.16×10^5	180	685	5905	3014	1542	0.51
S-CND	2	1.16×10^5	180	435	3750			
S-CND	3	4.22×10^4	180	136	3223			
S-CND	4	4.22×10^4	180	119	2820			
S-CND	5	4.22×10^4	180	136	3223			
S-CND	6	4.22×10^4	180	129	3057			
S-CND	7	4.22×10^4	180	51	1209			

S-CND	8	2.59×10^5	180	240	927			
-------	---	--------------------	-----	-----	-----	--	--	--

Concentration determination by using the nanoparticle tracking analysis (NTA)

Nanoparticle tracking analysis (NTA) was performed with a NanoSight LM10 (Malvern Panalytical) operated with a 405 nm laser.

R-CND solution were diluted to $c_{\text{CND}_s} = 18 \mu\text{M}$ (calculations of Supplementary Methods I - “Concentration determination by CND counting with transmission electron microscopy”) and *S*-CNDs to $c_{\text{CND}_s} = 5.5 \mu\text{M}$. As can be observed in Supplementary Figure 5, only large aggregates were tracked by the system for both samples. The main population of CNDs is too small and scatters too weakly to be discernable with this technique. The CND concentrations (which are in fact aggregate concentrations) determined with NTA were $n_{\text{CND}_s} \approx 3.4 \times 10^7$ CNDs mL^{-1} for *R*-CNDs and $n_{\text{CND}_s} \approx 1.0 \times 10^7$ CNDs mL^{-1} for *S*-CNDs. This corresponds to concentrations in the femtomolar range, underlining that CNDs cannot be measured with NTA but also that the number of aggregates in the CND solutions seems negligible. Note that presence of agglomerates to a large extend can be ruled out by the FCS measurements shown in Supplementary Methods III.

Supplementary Figure 5. Nanoparticle tracking analysis of *R*- and *S*- CNDs. Apparent hydrodynamic diameters d_h of tracked CNDs (aggregates) for *R*-CNDs (green line) and *S*-CNDs (red line).

Are there implications of this chirality issue for janus-like particles of architecture of alternating stripe-like surface feature?

⇒ One could introduce chirality to NPs for example by making janus-like particles of by appropriate surface patterns. However, such approach would work best on bigger particles. Our particles with around 2 nm diameter would be too small to allow for controlled

patterning/structuring of their surface. For bigger particles the problems of concentration determination would not apply. Also, the most convenient particles for patterning/structuring of their surface are typically metal nanoparticles, and then again concentrations could be determined via ICP-MS for example. Thus, we decided not to mention these cases in this article, as they would deviate from the main message.

Overall recommendation: manuscript not suitable for publication. May be suitable after major revisions.

⇒ We have extensively revised the manuscript and we hope we could rule out the concerns of the reviewer with the revised version.

Reviewer #2:

In the current study, carbon nanodots (CNDs) without/with opposite chirality (R-CNDs and S-CNDs) were synthesized, followed by detailed analysis about the comparison of the concentration of both types of CNDs. The authors also studied protein adsorption quantification in 3 typical proteins and demonstrated that CNDs are very weak binders of those proteins. There was only significantly less HSA adsorption on S-CNDs than on R-CNDs and this difference was associated with different cellular behaviors in THP-1 derived macrophages, which endocytose S-CNDs to a significantly higher extent than R-CNDs. In general, the current study is dealing a critical aspect of physicochemical properties of chirality of nanoparticles and the authors identified some interesting observations due to the differential absorption of chiral molecules and this directly related to different cellular uptake. There are some key issues the authors may need to further address to improve the current manuscript.

1. The overall biological significance of the study needs to further improved and there is no clear purpose why cellular uptake investigated and what is the matter.

⇒ We agree with the reviewer and in fact this goes in the same direction as the criticism by reviewer #1 and reviewer #3. Influence of chirality on particle uptake has been reported by others and we have also referenced this in the introduction. The main point of our study was to find out if such dependence can also be detected with ultrasmall organic nanoparticles (the CNDs), whose concentration determination is complicated. In cases where concentration can only be detected within a certain error, the biological effect must be bigger than this error in order to make a sound statement. We have shown how such error analysis can be performed and to which extent uptake is above such error. This is the significance of our study. We have tried to point this out better in our manuscript.

Concerning the biological relevance of controlling particle uptake, we believe that having handles to tune the uptake efficiency by physicochemical properties outlines possibilities for better achieving envisaged biodistributions. We have added a short section in the discussion:

The data shown here demonstrate that also under most stringent considerations of errors in the concentrations determination of CNDs, it has been shown that chirality may affect the protein corona formation and *in vitro* cellular uptake of CNDs to an extent of >20%. It thus can be

speculated that this difference would also influence the *in vivo* interaction of CNDs. Chirality itself does not influence the most important physicochemical properties of CNDs, such as fluorescence and colloidal properties. By using CNDs of different chirality thus different biodistributions of otherwise identical CNDs might be obtained.

2. The authors studied only 3 typical proteins and then generated the conclusion. It is recommended to study the serum proteins and if the same less protein binding is observed then the conclusion is more reliable.

⇒ The reviewer is correct, we have carried out study on 3 typical serum proteins. With our method, i.e. FCS we would not be able to quantify protein adsorption with a mixture of serum proteins or full serum. Our analysis is based on size changes upon protein adsorption, but is not sensitive to which types of proteins from a mixture would be adsorbed. To identify adsorbed proteins typically mass spectroscopy is performed, which is a complementary method. Our method on the other hand is fully *in situ*, without the need to purify the NP sample before measurements as it would be necessary for mass spectroscopy quantification. In particular for the very small particles, like the ones employed here, only a few proteins may bind and such purification could introduce a significant error, which is why we decided to carry out *in situ* measurements by FCS on selected protein solutions. Protein adsorption can be also quantified in blood (plasma) via size measurements, but without knowing which proteins have adsorbed. We have added the following statement:

We want to mention that our data refer only to three selected serum proteins. As with our method, i.e. FCS, we only detect changes in the hydrodynamic diameter upon protein adsorption, upon exposure to blood we would not be able to tell which proteins had adsorbed and caused the increase in size of the NPs. In order to detail the composition of the protein corona typically mass spectroscopy analysis is performed {Johnston, 2017 #34879}. However, for such measurements first unbound excess proteins have to be removed. For the small CNDs as investigated here to which only few proteins can weakly bind, such purification may significantly change the protein composition left on the NP surface {del_Pino, 2014 #25251}. In contrast, diffusion measurements with FCS are performed *in situ*, without the need for purification. While such measurements do not allow for telling the composition of the adsorbed protein corona, and thus are best carried out in different solutions containing only one type of model protein, diffusion measurements can still verify protein corona formation in blood {Carril, 2017 #34851}.

3. The authors studied a few cell lines, one of which is macrophage-like cells. It is more convincing to study other macrophages, such as primary macrophages to confirm the observation.

⇒ With all respect, we do not see how measurements with another cell line would change the message of our study. The reviewer is completely correct that for predicting potential *in vivo* effects of the chirality of CNDs it would be better to work with primary cells. But this is not the goal of our study. *In vivo* effects of chirality on NP uptake have been reported by others, which is referenced in the introduction. Our goal was to find out if influence of chirality may also be confirmed with NPs whose concentration is complicated to quantify. We have shown

that for one cell line, the macrophages, the biological effect is bigger than errors in concentration determination. For another cell line, i.e. HeLa cells, the effect was smaller than the error in concentration determination. Thus, we have examples for both cases. Any other cell would fall in the category of one of those two cases and thus we feel we have demonstrated our case well enough.

4. As the authors mentioned that several physicochemical properties entangled together, the authors may discuss more about their results in this context.

⇒ We have not pointed this out better in the conclusions. In fact, data of the N-CNDs were not related to their achirality, as these NPs showed signs of colloidal instability and thus effects were more likely related to colloidal instability than to chirality. We have added this section:

Several physicochemical properties may be entangled {Xu, 2018 #35515}, and many time effects may not be due to a primary physicochemical parameter (such as e.g. chirality), but due to colloidal stability (as here in the case of the N-CNDs). In fact, we have shown here by FCS measurements that for the *R*- and *S*-CNDs chirality does not affect colloidal stability and only because for this case entanglement of chirality and colloidal stability was ruled out differences in biological effects can be related to chirality as physicochemical parameter. For the N-CNDs there was an effect on colloidal stability and thus differences in their biological effects cannot be related to their non-chiral nature.

5. Furthermore, other pairs of chiral counterparts of NPs are studied? What the general pattern is expected?

⇒ We see the following cases, which have been also mentioned in the manuscript. For molecules/particles with precise molecular structure concentration determination is not an issue. For particles without precise molecular structure in case they are big enough conventional methods for concentration determination by particle counting can be applied. For both of these cases there are reports about the influence of chirality on their biological impact, which we have also referenced. In both cases in fact it has been shown that chirality can influence interaction with biology. In our case we have investigated the special case of very small NPs, which however don't have a precise molecular structure. Also here we showed that, depending on the type of protein and cells, there may be an effect of chirality on protein adsorption and particle uptake by cells. Thus, the general pattern would be that chirality may influence interaction of particles/molecules with biology over a significant size range. Examples are quoted in the introduction.

Reviewer #3:

In this paper, carbon nanodots (CNDs) with opposite chirality and achiral nanodots were prepared and their interactions with proteins and cells were studied. Protein adsorption quantification and internalization pathways were investigated painstakingly. This is an interesting research in chiral nanostructures. There are several points need to be clarified.

1. Some related references were missing. For example: (1) *Angewandte Chemie International Edition*, 2021, 10.1002/anie.202101609. (2) *Nature Communications* 2017, 8, 2007.

⇒ We are grateful to the reviewer for making us aware of these highly relevant references, which we oversaw. We have quoted them now.

2. Sub-Title ‘Co-localization studies of mitochondria of lysosome and CNDs’ in SI, there is something wrong with the expression.

⇒ We thank the reviewer for having found this typographic error, which we have fixed now.

3. CNDs used in this study were synthesized according to previously reported methods. I think the basic physicochemical characteristic data should still be provided in supporting information, such as atomic force microscope images, circular dichroism spectra, Fourier-transformed infrared spectroscopy (FT-IR) and X-ray photoelectron spectroscopy (XPS) and so on.

⇒ We agree with the reviewer. We have added these data which were exclusively measured for this present study in the Supporting Information. There is nothing new, the data agree well with the previously published characterization data.

Additions to the Supporting Information:

Supplementary Figure 28. AFM of R-CNDs. Tapping mode AFM ($5.0 \times 5.0 \mu\text{m}$) from a drop-casted aqueous solution of R-CNDs on a mica substrate (scale bar, $1 \mu\text{m}$). The inset is the height profile $h(d)$ along the dashed line.

Supplementary Figure 29. AFM of S-CNDs. Tapping mode AFM ($5.0 \times 5.0 \mu\text{m}$) from a drop-casted aqueous solution of S-CNDs on a mica substrate (scale bar, $1 \mu\text{m}$). The inset is the height profile $h(d)$ along the dashed line.

Supplementary Figure 30. TEM of *R*-CNDs. TEM images from dried drop-casted aqueous solution of *R*-CNDs on TEM grids.

Supplementary Figure 31. TEM of S-CNDs. TEM images from dried drop-casted aqueous solution of S-CNDs on TEM grids.

Other physicochemical characterization data

Additional standard characterization of the CNDs is provided in form of Fourier-transform Infrared (FT-IR) spectra (KBr), shown in Supplementary Figure 2, electronic circular dichroism (ECD) spectra shown in Supplementary Figure 1 and X-ray photoemission spectroscopy (XPS) shown in Supplementary Figure 3. FT-IR spectra were recorded on a Perkin Elmer 2000 spectrometer. ECD spectra were measured on a Jasco J-815. XPS spectra were measured on a SPECS Sage HR 100 spectrometer.

Supplementary Figure 1. (Chiro)optical characterization of *R*- and *S*-CNDs in Milli-Q water. Electronic circular dichroism (ECD) spectra of *R*-CNDs (black line) and *S*-CNDs (red line) in water at 298 K. The results are in agreement with our previous work.

Supplementary Figure 2. FT-IR spectra of *R*- and *S*-CNDs.(a) *R*-CNDs. (b) *S*-CNDs. The results are in agreement with our previous work.

Supplementary Figure 3. XPS of *R*- and *S*-CNDs. XPS survey of (a) *R*-CNDs and (b) *S*-CNDs showing the C1s, N1s and O1s; deconvoluted C1s spectra of (c) *R*-CNDs and (d) *S*-CNDs; deconvoluted N1s spectra of (e) *R*-CNDs and (f) *S*-CNDs. The results are in agreement with our previous work.

4. In addition, the potential use of chiral CNDs *in vivo* should be discussed.

⇒ We have added a short section in the discussion:

The data shown here demonstrate that also under most stringent considerations of errors in the concentrations determination of CNDs, it has been shown that chirality may affect the protein corona formation and *in vitro* cellular uptake of CNDs to an extent of >20%. It thus can be speculated that this difference would also influence the *in vivo* interaction of CNDs. Chirality itself does not influence the most important physicochemical properties of CNDs, such as fluorescence and colloidal properties. By using CNDs of different chirality thus different biodistributions of otherwise identical CNDs might be obtained.

REVIEWERS' COMMENTS

Reviewer #1 (Remarks to the Author):

The authors have adequately and in sufficient detail addressed the issues/concerns raised in my review. I recommend publication

Reviewer #2 (Remarks to the Author):

The authors have addressed most issues raised by the reviewers. Since this is a quite quantitative and fundamental study to compare the toxicity of nanoparticles and their following biological effects, I have no further questions at the current stage. Congratulations for the authors to produce a nice piece of work.

Reviewer #3 (Remarks to the Author):

The authors have adequately addressed the concerns raised by this reviewer.